ecology/biogeography/biomathematics

ecological community dynamics, plant diversity, species coexistence, biogeographic patterns, null hypotheses testing, stochastic Markov processes in continuous time

**Authors for correspondence:**
José A. Capitán
e-mail: ja.capitan@upm.es
David Alonso
e-mail: dalonso@ceab.csic.es

# A signal of competitive dominance in mid-latitude herbaceous plant communities

José A. Capitán[1,2], Sara Cuenda[3], Alejandro Ordóñez[4] and David Alonso[2]

[1]Complex Systems Group, Department of Applied Mathematics, Universidad Politécnica de Madrid, Av. Juan de Herrera, 6, 28040 Madrid, Spain
[2]Theoretical and Computational Ecology, Center for Advanced Studies (CEAB-CSIC), C. Accés Cala St. Francesc 14, 17300 Blanes, Catalonia, Spain
[3]Facultad de Ciencias Económicas y Empresariales, Depto. Análisis Económico: Economía Cuantitativa, C. Francisco Tomás y Valiente 5, Universidad Autónoma de Madrid, 28049 Madrid, Spain
[4]Department of Bioscience, Aarhus University, Aarhus, Ny Munkegade 114, 8000 Aarhus C, Denmark

 JAC, 0000-0002-6245-0088; DA, 0000-0002-8888-1644

Understanding the main determinants of species coexistence across space and time is a central question in ecology. However, ecologists still know little about the scales and conditions at which biotic interactions matter and how these interact with the environment to structure species assemblages. Here we use recent theoretical developments to analyse plant distribution and trait data across Europe and find that plant height clustering is related to both evapotranspiration (ET) and gross primary productivity. This clustering is a signal of interspecies competition between plants, which is most evident in mid-latitude ecoregions, where conditions for growth (reflected in actual ET rates and gross primary productivities) are optimal. Away from this optimum, climate severity probably overrides the effect of competition, or other interactions become increasingly important. Our approach bridges the gap between species-rich competition theories and large-scale species distribution data analysis.

## 1. Introduction

Biodiversity theory in community ecology heavily relies on the pioneering work of Volterra [1] and Lotka [2]. These authors provided a general framework to mathematically describe the interacting dynamics of natural populations. These seminal

ideas have been extensively developed mostly focusing on the analysis of simple ecological communities. For instance, Chesson and colleagues [3–6] introduce a general framework—the *modern coexistence theory* for competitive communities—to understand species coexistence in natural communities based on pairwise species differences and their interplay to determine effective competitive (biotic) interactions. According to this framework, the balance between stabilizing trait differences and species dominance among competitors is crucial to understand species coexistence. In communities driven by fitness differences, species turn out to be clustered around similar trait values selected through competitive dominance. However, trait clustering may arise through two radically different mechanisms. Independent adaptation of non-interacting species to the same environmental conditions can lead to trait clustering. The alternative explanation would say that competitive interactions leading to fitness equalization end up producing more similar species, with, therefore, more similar traits. Therefore, trait clustering may be interpreted as a fingerprint of competition even in the absence of environmental filtering [6,7]. These ideas have proved challenging to apply to large ecological communities. Rather than focusing on whether (or not) and why ecological similarity among species should arise (or not) in natural communities, Hubbell and colleagues assumed ecological equivalence as a first principle and studied the consequences of this assumption for species coexistence and community-level patterns in species-rich systems [8–10]. Other authors, building on May's seminal work [11], have used a random matrix approach to advance understanding on species coexistence in large communities through mathematical analysis [12–15]. Statistical physics has also helped to understand how pairwise species interactions scale up to determine the type of dynamic stability and potential species coexistence in species-rich large systems [16].

Although the role of local interactions at determining large-scale diversity patterns is still controversial [17], community ecology lacks a comprehensive theoretical framework able to explore quantitatively to what extent the role of biotic, species-to-species interactions is relevant to determine species composition and diversity across large spatial scales. Empirical studies, while they may be able to independently assess environmental stress and species competitive abilities, are often limited to small community sizes [18] or restricted to single habitats [19]. Very few studies have explored the idea of competition as a driver of community assembly across biogeographic regions [20,21]. Here we attempted a continent-wide macro-ecological study of species assemblage patterns based on theoretical predictions from a trait-driven theory of competitive dominance, using extensions of a type of Lotka–Volterra models. Our theory applies to large ecological communities at large geographical scales where species can be ranked in their competitive ability according to certain species trait values [22].

Light and water availability (figure 1) impose significant limitations on gross primary productivity which is reflected in actual evapotranspiration (ET) rates [23]. These two resources vary at regional scales, placing strong, sometimes opposing constraints on how tall a plant can grow. Plant height is a fundamental trait that reflects the ability of the individual to optimize its own growth within its local biotic environment and regional physical constraints (see [24,25] and references therein). How plant height adapts to these opposing constraints has been studied in trees [26–28] and herbaceous plants [29,30]. Here we analysed presence–absence matrices of floral herbaceous taxa across different European ecoregions to determine if competitive ability (reflected in maximum stem height) could help explain assemblage patterns at local scales across gradients of relevant environmental factors such as ET. We examined how well observed plant assemblages at macro-ecological scales match theoretical predictions generated by a synthetic, stochastic framework of community assembly [31–35], which we described in full detail in Capitán *et al.* [22]. By assuming that competition between heterospecifics is driven by signed height differences, we found a significant positive correlation between the degree of clustering and actual ET rates (or gross primary productivity, GPP). Across Europe, actual ET (and GPP) is lower at more southern latitudes (due to reduced precipitation levels) as well as at more northern latitudes (due to colder temperatures and low levels of sunlight). Herbaceous plant height clustering is significant only over a latitudinal band where environmental constraints to plant growth are weaker, which suggests that the signature of competitive dominance can only be detected in the assemblage patterns of mid-latitude ecoregions.

## 2. Theoretical predictions

Recently, we presented a stochastic framework of community assembly [22]. This framework provides a stochastic extension of Lotka–Volterra competition models. While other extensions consider only symmetric competition on theoretical grounds [33], our approach relates specifically measurable

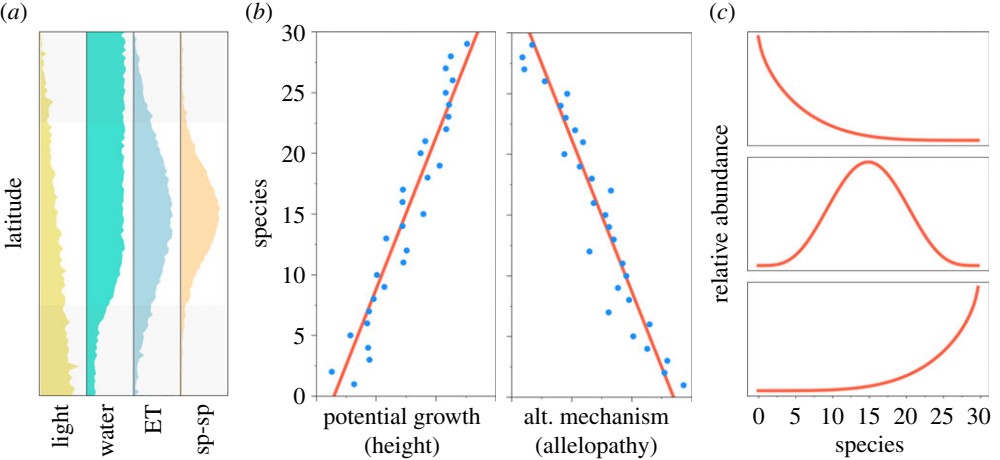

**Figure 1.** Conceptual framework for maximum height resulting from a trade-off between investing energy either in potential growth, or in any other alternative, non-size-related strategy. In panel *a*, we illustrate latitudinal patterns of potential light and water availability. The latitudinal gradient of actual evapotranspiration (ET) is also shown along with the expected role of biotic interactions in determining community dynamics. At middle-range latitudes, we expect competitive hierarchies to be at their maximum due to a greater relative role of species interactions. Panel *b* shows how the trade-off between potential growth and any alternative mechanism not related to size can be included in a spatially explicit model: species that are either good at growing taller or in investing energy in allelopathy remain short, but cause incremental death of their heterospecific neighbours. As an outcome of this trade-off, the model predicts the dominance of taller, mid-sized or shorter plants at stationarity (panel *c*).

species traits and competitive dominance. In order to make this contribution self-contained, we first provide a summary of the main predictions from our theory [22]. We developed first a single-trait driven, spatially implicit species-competition model. Then, we extended this model into space and incorporated a second trait controlling species competition. Both models together provided us with rich predictions that can be tested with appropriate species assembly data. Below we summarize these predictions.

## 2.1. Two predictions from the implicit model

### 2.1.1. Species coexistence decays with competition intensity

Recent theoretical approaches have focused on predicting analytically the expected fraction of species that survive in competitive scenarios [15]. A spatially implicit model of Lotka–Volterra type [22] allowed us to predict on average how many species are expected to survive as a function of mean competitive strengths. We observed that the fraction of extant species $p_c$, which we called 'coexistence probability', decays with the average competitive strength $\langle \rho \rangle$ as a power law above a certain threshold in competition, and curves for different pool sizes $S$ can be collapsed into the same curve following the mathematical dependence

$$p_c \sim (\langle \rho \rangle S)^{-\gamma}, \tag{2.1}$$

which was observed numerically and justified analytically [22]. We showed that the exponent $\gamma$ is controlled by the immigration rate $\mu$. This is the first prediction of the spatially implicit model.

### 2.1.2. Species clustering under competitive dominance

In order to explore the significance of competitive dominance in empirical communities, we applied first randomization tests to model communities. In this way, we established a second prediction for this model. Null models for community assembly [36–38] compare the properties of actual communities against random samples of the same size extracted from a species pool (observed diversity at the ecoregion level). This approach assumes that realized communities are built up through the independent arrival of equivalent species from the pool [39–41] regardless of species preferences for

particular environments or species interactions. Our randomization tests were based on a single statistic, the competitive strength averaged over species present in realized model communities, which were then compared to random samples of the same size drawn from the species pool. The null hypothesis (i.e. empirical communities are built as random assemblages from the ecoregion) can be rejected in both sides of the distribution, implying signals of 'significant trait overdispersion' (clustering) if average trait differences are larger (smaller) than expected at random. In the low immigration regime, the model predicts a significant signal of clustering. This regime is characterized by a low non-dimensional immigration rate ($\lambda = \mu/(\alpha K)$ much lower than 0)—here $\alpha$ stands for the average species growth rate in isolation, and $K$ is the carrying capacity of the environment.

## 2.2. Two predictions from the explicit model

The spatially explicit model incorporates a trade-off between potential growth and the production of allelopathic compounds. This alternative mechanism would allow shorter individuals to overcome being outcompeted by taller plants [22]. Our model explores how taller species, which are better competitors for light, and shorter ones, which allocate more energy in allelopathic compounds, coexist in a single interacting community on a given area (figure 1).

### 2.2.1. Competitive dominance may select for shorter plants

Height hierarchies alone, as assumed in our spatially implicit model, lead to the selection of taller plants in species assemblages. In the more realistic spatially explicit model, species processes take place on a lattice where locally taller plants grow faster than neighbours because they are less shaded, but in the presence of heterospecific neighbours, they are also more prone to die. Computer simulations show that the balance of these two mechanisms can end up selecting plant sizes characterized by an optimal potential height that can be either shifted toward lower or higher values depending on the choice of model parameters. This is the first prediction of the spatially explicit model: species abundance distributions are not necessarily biased towards taller individuals, and they can peak at species at intermediate or even shorter heights. In any case, and consistently, in this more complex scenario, a balance between the gains of potential growth and the gains of energy allocation in allelopathy (as an example of a non-size-related, alternative mechanism) may result in a selection for plants exhibiting significant height clustering at stationarity.

### 2.2.2. Clustering patterns hold across aggregation scales

A second result that can be derived from the spatially explicit model is related to the persistence of trait clustering when species are aggregated over spatial scales larger than local interaction distances. Our spatially explicit model can help explain why clustering patterns persist over large scales. The distributions of species within a region may reveal more information about the underlying assembly processes than the co-occurrence of species at any given location [17]. As species are aggregated over lattice cells of increasing size, clustering patterns hold even at scales much larger than local interaction distances. The model predicts consistent clustering patterns regardless of the aggregation scale used to define species communities. This was the second prediction, derived and carefully analysed in Capitán *et al.* [22], from our spatially explicit model.

# 3. Materials and methods

Plant community data were drawn from Atlas Florae Europaeae [42]. The distribution of flora is geographically described using equally sized grid cells (approximately $50 \times 50$ km) based on the Universal Transverse Mercator projection and the Military Grid Reference System, see figure 2. Each cell was assigned to a dominant habitat type based on the WWF Biomes of the World classification [43], which defines different ecoregions, i.e. geographically distinct assemblages of species subject to similar environmental conditions. We consider each cell in an ecoregion to represent a species aggregation.

Each herbaceous species in an ecoregion was characterized by its maximum stem height $H$, an eco-morphological trait that relates to several critical functional strategies among plants [44]. It represents an optimal trade-off between the gains of accessing light [26,27], water and nutrient transport from soil

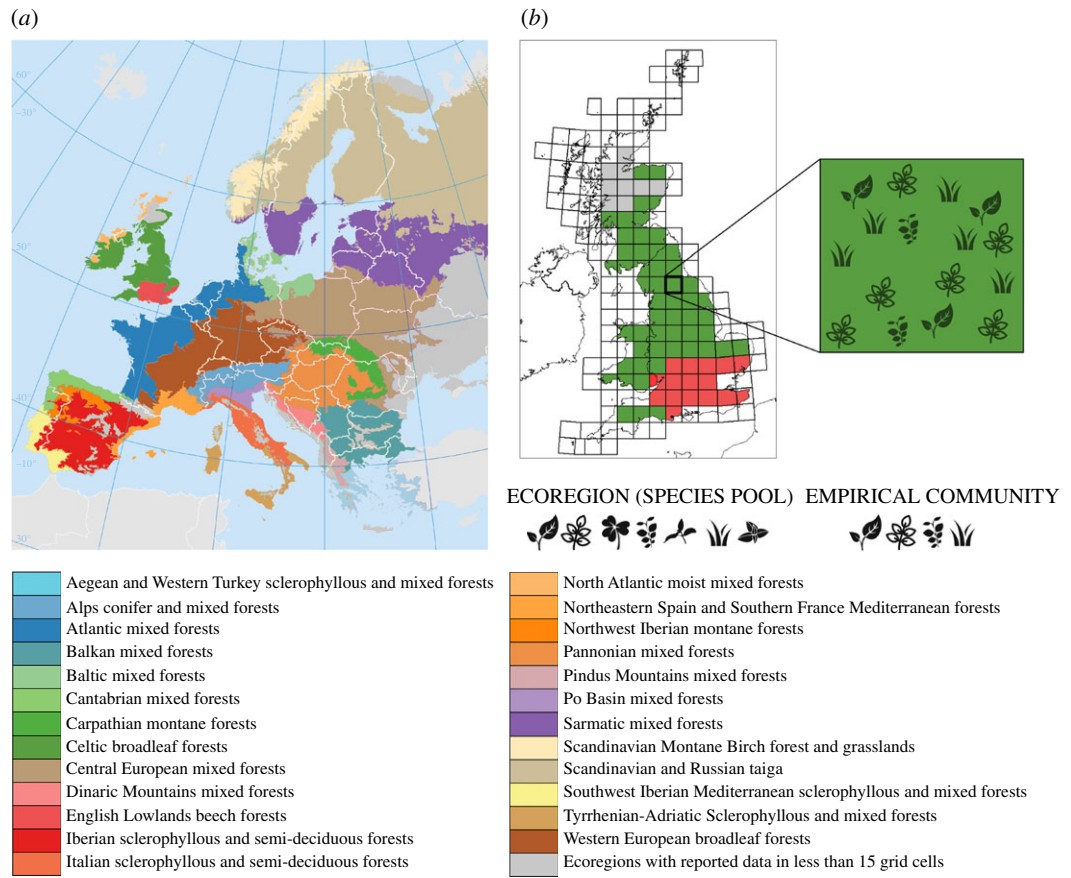

**Figure 2.** Geographical description of plant data across European ecoregions. (*a*) Twenty-five different habitats covering most of Europe are shown in the map and listed below. Ecoregions are regarded as a pool comprising all plant species observed in that region. (*b*) The Military Grid Reference System divides ecoregions in grid cells, each one considered as an assemblage formed by a species sample of the pool.

[28,45], and additional constraints posed by the local biotic environment of each individual plant, such as competition, facilitation or herbivory.

Mean height values were obtained from the LEDA database [46] for as many species as there were available in the database. Missing values were taken from [47] or inferred using a multivariate imputation by chained equations (MICE) approach [48] together with a predictive mean matching algorithm based on other available traits (leaf and seed traits), genus and growth forms as predictors. Based on plant growth forms, 2610 herbaceous species (aquatic, herbs or graminoid) were considered in this work.

Maximum stem height values spanned several orders of magnitude, so we used a log-transformed variable ($h = \log H$) to measure species differences (using non-transformed heights yielded comparable results, here not shown). The values of $h$ were standardized within ecoregions as $t = (h - h_{min})/(h_{max} - h_{min})$ so that $0 \leq t \leq 1$.

For all the species reported in an ecoregion, we formed an empirical competition matrix with pairwise $\rho_{ij}$ signed height differences $\rho_{ij} = \hat{\rho}(t_j - t_i)$, where $t_i$ are height values standardized across ecoregions and sorted in increasing order. The advantage of having these values represent trait differences between pairs of species is that any trend in competitive strengths can be immediately translated into patterns of functional trait clustering or overdispersion. As suggested in Capitán *et al.* [22], we calculated the average pairwise competitive strength as

$$\langle \rho \rangle = \frac{2}{S(S-1)} \sum_{i=1}^{S} \sum_{j=i+1}^{S} |\rho_{ij}|, \tag{3.1}$$

where $S$ stands for ecoregion richness.

In an ecoregion with richness $S$, a number $s_k \leq S$ of species will form a species assemblage at cell $k$. The coexistence probability was calculated from data as the average fraction of species that survive per cell

$$p_c = \frac{\langle s \rangle}{S} = \frac{1}{SN_C} \sum_{k=1}^{N_C} s_k,$$ (3.2)

with $N_C$ representing the number of cells in the ecoregion. This quantity, together with the distribution of trait differences in cells, was used to compare model predictions with real data.

ET maps were obtained from data estimated through remote sensing. ET data at different spatial and temporal resolutions were taken from the MODIS Global Evapotranspiration Project (MOD17), a part of the NASA/EOS project to estimate terrestrial ET from land masses by using satellite remote sensing information (http://www.ntsg.umt.edu/project/modis/mod17.php). Available datasets estimate ET using the improved algorithm by Mu *et al.* [49].

## 3.1. Randomization tests

Following Triadó-Margarit *et al.* [50], our randomization tests applied to empirical communities were based on the average competitive strength observed in a cell $C$ formed by $s$ species,

$$\langle \rho \rangle_C = \frac{2}{s(s-1)} \sum_{i=1}^{s} \sum_{j=i+1}^{s} |\rho_{ij}^C|,$$ (3.3)

where $(\rho_{ij}^C)$ is the submatrix of the ecoregion competition matrix restricted to the species present in the cell. Compared to ecoregion samples, the lower (higher) the empirical community average $\langle \rho \rangle_C$ is, the higher (lower) is the degree of species clustering in the cell. For each cell, we calculated the probability $p = \Pr(\langle \rho \rangle_Q \leq \langle \rho \rangle_C)$ that the competition average $\langle \rho \rangle_Q$ randomly sampled from the pool is smaller than the empirical average. At a 5% significance level, if $p > 0.95$ the empirical competition average is significantly larger than the average measured for random pool samples, which implies that average trait differences in realized communities are larger than would be expected at random. On the other hand, if $p < 0.05$, observed trait differences are significantly smaller than would be expected at random. Therefore, if $p > 0.95$, the community exhibits 'significant trait overdispersion', whereas if $p < 0.05$, there is evidence for 'significant trait clustering' in the observed species assemblage.

# 4. Results

If larger plants capture more resources, evolution should favour investment in potential growth (maximum height) as a competitive mechanism. However, investment in alternative mechanisms, such as allelopathy, may help smaller plants stave off competitors, reducing local heterospecific plant cover and giving them a competitive advantage over potentially taller plant species. As a consequence, the maximum species stem height can be regarded as the outcome of an evolutionary game [30] that balances opposing constraints, both physical [24,51] and biotic [26,27]. To explore these opposing constraints, we analysed plant data in the light of the two community assembly models. The first one is a spatially implicit model of Lotka–Volterra type, and the second one is a straightforward spatially explicit extension including height-driven competition and allelopathic effects. Both have been carefully defined and studied in Capitán *et al.* [22].

## 4.1. Two predictions from the implicit model tested against data

### 4.1.1. Species coexistence decays with competition intensity

The collapse of curves predicted by equation (2.1) helps eliminate the variability in $S$, so that empirical coexistence probabilities, which arise from different ecoregion sizes, can be fitted together (figure 3). Confirming the first prediction of the spatially implicit model, we found a significant correlation between the probability of coexistence and the scaled competitive overlap based on empirical data (figure 3), indicating that a model driven solely by dominant competitive interactions reliably predicts the average richness of plant communities across ecoregions. In addition, this theoretical prediction allowed an indirect estimation of the relative importance $\hat{\rho}$ of average inter- versus intra-specific effects: the average ratio of inter- to intra-specific competition strength is about 4% (see electronic supplementary material, section A for details on the estimation procedure).

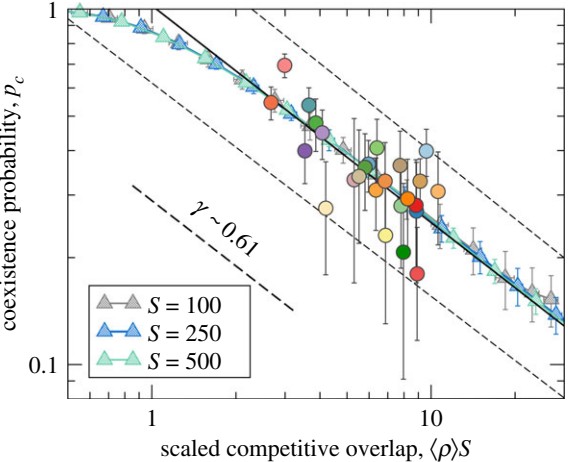

**Figure 3.** The implicit model predicts a power-law decay regardless of the ecoregion size $S$, which permits fitting a power law to data ($r^2 = 0.51$, $p < 10^{-3}$, 95% confidence lines are shown). In order to match the empirical exponent $\gamma$ we need to choose the immigration rate $\mu = 5$, the net growth rate $\alpha = 50$ and the carrying capacity $K = 1000$. To match the starting point of the decay we need to set $\hat{\rho} = 0.04$ in the calculation of $\rho_{ij}$. For completeness, we have reproduced here model expectations (triangles) for different pool sizes. Data colours match ecoregion codes in figure 2.

### 4.1.2. Species clustering under competitive dominance

As a second prediction, the implicit model predicts species clustering under competitive dominance under certain parameter regime. High levels of trait clustering are only found for low immigration rates and high carrying capacity values. Importantly, this is the parameter regime that seems to precisely emerge from the data. In Capitán *et al.* [22], we derived a deterministic prediction for the exponent, $\gamma = 1$, under no immigration, which does not match the one obtained from data ($\gamma = 0.61$). As we showed [22], it is a non-zero (but small) value of the immigration rate that determines the value of the power-law exponent $\gamma$ that becomes lower than 1 in the case of non-zero immigration. Indeed, for a realistic fit in figure 3, the exponent of the empirical power law is obtained for $\mu/\alpha \sim 0.1$ individuals per generation. Since plant communities operate in a low-immigration regime, the non-dimensional immigration rate $\lambda = \mu/(\alpha K)$ must satisfy $\lambda = 0.1/K \ll 1$, hence the carrying capacity must be large. Indeed, in the same parameter regime where empirical coexistence probabilities are best predicted, that is, low immigration rate and high carrying capacity, the implicit model predicts a significant degree of species clustering (see fig. 3 in Capitán *et al.* [22]).

Testing this second prediction against empirical observations yields a mixed picture. We calculated $p$-values for randomization tests applied to every cell in each ecoregion, which represent the empirical distribution of $p$-values (figure 4). At the parameter values that make plant data consistent with the first prediction, the spatially implicit model predicts significant trait clustering. We observe that some ecoregions are consistent with this theoretical expectation. However, other ecoregions clearly do not comply with this prediction. In addition, no ecoregion is consistent with trait overdispersion (figure 4). Selecting species in randomization tests according to species dispersal abilities portrays the same picture (results not shown).

## 4.2. Ecoregion clustering and actual evapotranspiration rates

We explored whether there is a geographic signal in the propensity of an ecoregion to exhibit clustering in maximum stem height. For a better quantification, we defined a clustering index $q$ for an ecoregion as the fraction of its cells that lie within the 5% range of significant clustering (randomization tests yield $p$-values smaller than 0.05 for those cells). An ecoregion for which significant clustering is found in most of its cells will tend to score high in the $q$ index. We examined how the clustering index varied across the continent in terms of the geographical location of ecoregion centroids as well as with actual ET (figure 5).

Water availability acts as a factor limiting plant growth at geographical scales (figure 1*a*). However, water has to be channelled up through stems and leaves for effective growth to take place. Therefore, at large geographic scales, growth primary productivity positively correlates with ET [23] (figure 5*d*).

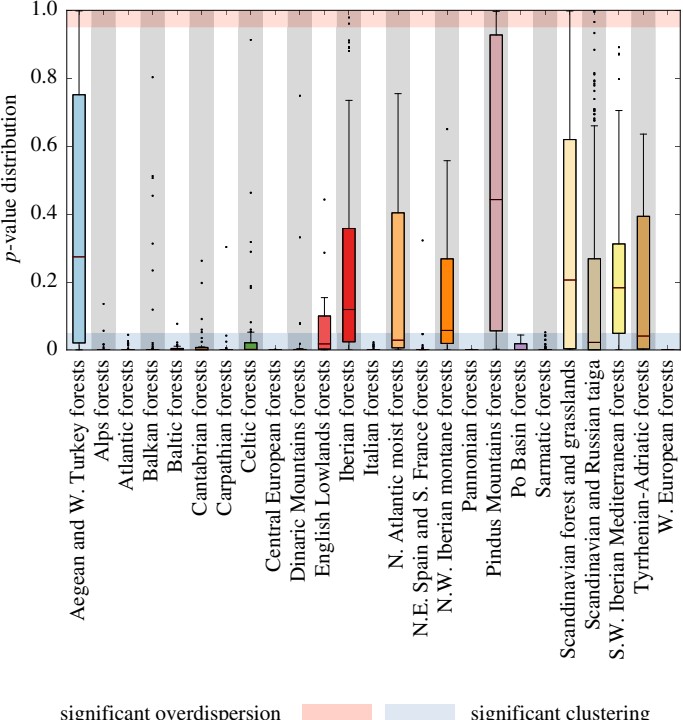

**Figure 4.** Empirical randomization tests. Over half of the ecoregions are consistent with model predictions as the distributions (Tukey boxplots) lie in the 5% range of significant clustering (Methods). We present here distributions of *p*-values across local communities in every ecoregion. Shaded areas would represent threshold *p*-values for two one-tailed tests where the hypothesis of trait clustering and overdispersion, in blue and pink, respectively, are represented on the same plot. Data colours in panels (*a,c*) match codes in figure 2.

Therefore, for a given region, mean annual ET is a reliable measure of environmental constraints on plant growth [23]. Figure 5*a,b* shows a clear latitudinal trend: there is an intermediate range of ecoregion latitudes where both clustering indices and ET are large, indicating that ET measures can robustly predict clustering indices (figure 5*c*). The same pattern can also be seen in the relation between mean relative height differences and actual ET across individual grid cells. The intensity of the clustering pattern increases with actual ET rates across Europe, not only at the ecoregional level (figure 5*c*), but also at the lower spatial scale of grid cells (see electronic supplementary material, figure C1). More importantly, since ET is a powerful proxy of environmental constraints on plant growth, this clustering in maximum stem height appears to be stronger at ecoregions less limited by environmental conditions. As environments become harsher and less optimal for plant growth, these clustering patterns disappear. This is particularly true for the severe climatic conditions characteristic in the Mediterranean (with erratic rainfall, limited water availability and drought), as well as of boreal zones (with low radiation incidence and cold temperatures). According to model predictions, the overall clustering patterns found at middle-range latitudes appear to be consistent with species competitive dominance shaping species height differences.

## 4.3. Two predictions from the explicit model tested against data

### 4.3.1. Competitive dominance may select for shorter plants

The spatially explicit model allows for either the dominance of tall, mid-sized or short plants, as a consequence of the trade-off between investment in either potential growth or alternative mechanisms other than growth (see fig. 5 in Capitán *et al.* [22]). We have tested whether taller or shorter plants are most commonly represented in ecoregions via the correlation of cell-averaged heights and ET (figure 6*a*), which shows a mixed picture. With few exceptions, mid-latitude ecoregions exhibit positive correlation (taller plants are selected in regions favouring plant growth), whereas negative dependencies are often observed in latitudinal extremes (figure 6*b*). Correlations are significant but, in

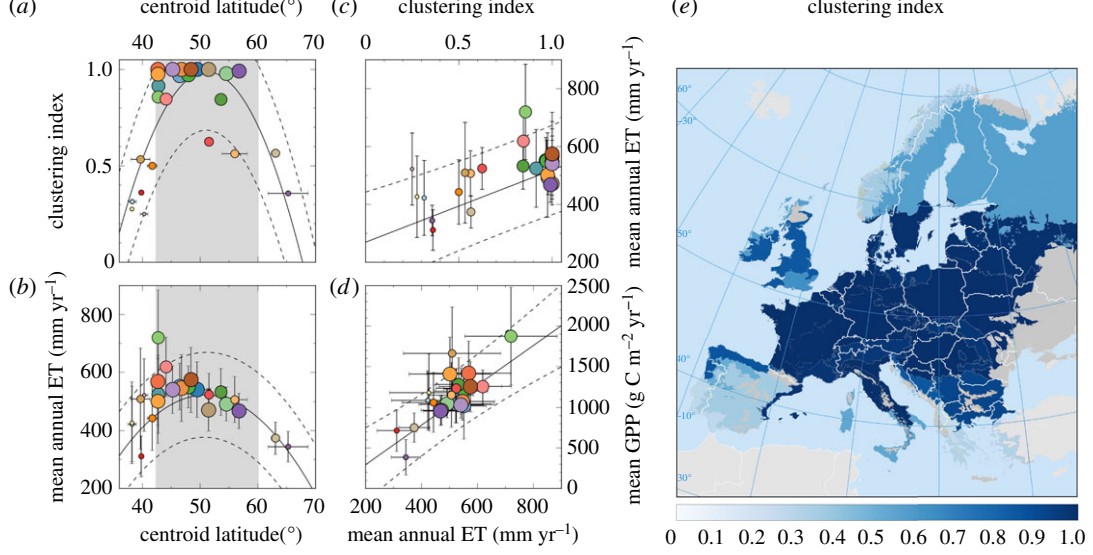

**Figure 5.** Linking height clustering to geographical and environmental variables. (*a*) Variation in the clustering index (*q*) with latitude (*φ*). Quadratic fit: $r^2 = 0.63$, $p < 10^{-3}$. (*b*) Latitudinal variation in mean annual actual ET data. Quadratic weighted regression: $r^2 = 0.63$, $p < 10^{-3}$. The shaded areas in panels (*a,b*) represent the latitudinal range for which the adjusted dependence $q(\varphi) \geq 0.7$, where both height clustering and ET are maximal. (*c*) Linear weighted regression for ET as a function of the clustering index; $r^2 = 0.49$, $p < 10^{-3}$. (*d*) Correlation between mean gross primary productivity (GPP) and mean annual ET; linear weighted fit: $r^2 = 0.73$, $p < 10^{-3}$. In the first four panels, the radius of each circle is proportional to the clustering index. Symbol colours refer to ecoregions (figure 2). All the fits show the 95% confidence bands. (*e*) Geographical distribution of clustering indices for ecoregions across Europe.

some cases, very weak. These results are consistent with our interpretation in terms of a signal of competitive dominance in mid-latitude ecoregions.

### 4.3.2. Clustering patterns hold across aggregation scales

Our spatially explicit model predicts the persistence of trait clustering as species are aggregated at larger spatial scales (much larger than the typical range of species interactions). This is important because real individual plants interact at much lower spatial scales (1–1000 ha) compared to the spatial resolution of our dataset (grid cell sizes about 50 km). To assess the robustness of our results, we further investigated the effect of aggregation scales on clustering patterns using plant data. In line with the spatially explicit model, the analysis of herbaceous plant communities from mid-latitude ecoregions reveals that our results are robust to both up- and down-scaling community sizes (figure 6*c*). Height clustering remains significant in a range of aggregated scales, and extrapolates to smaller areas (under a random placement hypothesis, communities of smaller sizes were built by randomly selecting a number of species as predicted by the empirical species–area relation, see electronic supplementary material, section B). We conclude that clustering patterns at large scales is an emerging pattern that can be interpreted as a signature of competitive dominance operating at much smaller spatial scales.

## 5. Discussion

In this work, we have tested predictions from a model of species-rich interacting communities under competitive dominance [22]. Our work is based on spatial and stochastic extensions of a type of Lotka–Volterra models where competitive dominance is linked to species traits [22]. This piece of theory was initially inspired by the competition-similarity paradigm [6]. We used macro-ecological trait data at large spatial scales [20] to show that, while potential ET decreases with latitude, actual ET peaks at intermediate latitudes, and is strongly associated with higher levels of trait clustering. Critically, actual ET is positively correlated with gross primary productivity (GPP) across terrestrial ecosystems (figure 5*d*) [23], which also peaks at intermediate latitudes across Europe. Consistently, our results were reproduced using GPP instead of ET, although both variables yield similar results. The

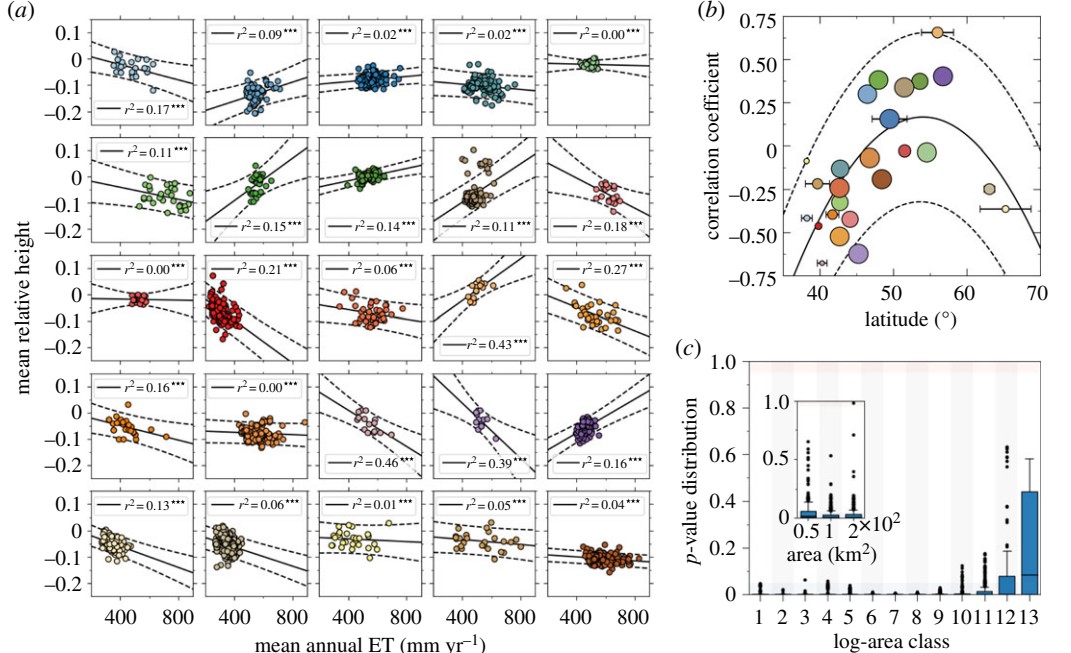

**Figure 6.** Two predictions of the explicit model tested against data. (*a*) Correlation of cell-averaged height (relative to ecoregion means) and mean annual ET by ecoregion (colours used for data match codes in figure 2). (*b*) Correlation coefficient obtained in (*a*) versus latitude. Circle radii are proportional to clustering indices. Observe that positive correlations tend to associate with high clustering index (with some exceptions) and middle-range latitude (quadratic fit: $r^2 = 0.44$, $p = 0.001$). (*c*) Clustering patterns of an ecoregion characterized by high clustering index (Atlantic mixed forests) were analysed at increasing aggregation scales. Communities were defined by increasingly aggregating contiguous $50 \times 50$ km cells. Below a critical aggregation scale (11th log-area bin, which corresponds to $10^5$ km$^2$), randomization tests show strong signals of clustering. The inset in (*c*) represents a down-scaling of randomization tests. Clustering patterns robustly persist at smaller spatial scales.

agreement of model predictions with plant community data can be interpreted as a signature of competitive dominance in empirical communities in the environmentally conducive middle-range latitudes. Significant height clustering would be the trace that competition leaves on community assembly pattern by filtering out subdominant species. If species tend to be similar in maximum stem heights at mid-latitudes, we suggest that this height equalization is a signature of competitive dominance. This mechanism would have played a key role in shaping local species assemblages through years and years of common eco-evolutionary history. This result does not necessarily mean that competition is the main driver of community assembly. It rather highlights the potential role of competitive dominance, along with other processes, in the assembly of herbaceous communities at intermediate latitudes. On the contrary, as environmental conditions get increasingly extreme, no significant clustering in plant height is observed. Although the interplay between facilitation and competition is far from simple [52], the harshness of extreme conditions probably override the effects of competition, and other processes such as species tolerances and facilitation [53,54] may be critical community drivers at climatic extremes.

Although we introduced our conceptual framework based on 'ideal plant growth conditions' (figure 1*a*), the patterns presented for light and water availability are not necessarily unimodal nor universal for all plant species. In general, many herbaceous plants grow efficiently when water availability is high, and temperatures are not extremely low. We acknowledge that there are exceptions to this rule. For example, environments that are too wet can lead plants to drown if their roots are saturated, which can cause early mortality and fast turnover (due to fungal infections, for instance). Likewise, high night-time temperatures can lead to increases in respiration rates, thereby reducing overall growth. Many of these relationships are discussed in Lambers & Oliveira [55]. Climatic drivers can induce a variety of effects on plant growth different from the generic trend we used here to frame our contribution.

Throughout this work, species assemblages within each grid cell (approximately $50 \times 50$ km) have been defined as distinct communities. Current consensus about the concept of ecological community emphasizes the importance of biotic interactions. An ecological community is defined as a set of species that live in the same area and can potentially interact [56]. In spite of the size and

heterogeneity within each grid cell at the $50 \times 50$ km spatial scale, cells are much smaller than the ecoregion they belong to, and are, of course, much more homogeneous, both in species composition and in environment, than the ecoregion itself. Therefore, in principle, grid cells could be regarded as communities in an operational and relative sense. In addition, we assumed that the European Flora database represents species composition at a steady state, that is, we examined the stationary patterns resulting from eco-evolutionary processes associated to long time scales. Although real individual plants interact at much lower spatial scales, two species from the same ecoregion will eventually interact within a grid cell given enough time. The larger the temporal scale, the larger is the area where two species will have a chance to interact through generations and repeated dispersal events. The scale at which a set of local communities reveal information about underlying assembly processes is very often the regional scale [57–59], which has led to the 'regional community concept' [17,60].

It is important to make a clear distinction between actual plant size and the species-level trait, 'maximum stem height'. While a species-level trait is shaped by evolutionary constraints at longer temporal scales, actual plant size is determined by a host of contingent ecological constraints operating over shorter temporal scales. Although there is a large body of theory and experiments positively co-relating actual plant size and individual plant competition ability [61,62], there has been considerably less attention paid to the evolutionary establishment of functional trade-offs between different species-level traits [63,64]. The common wisdom that competition favours taller plants may not always hold (for instance, in low-nutrient, competition-intensive, undisturbed habitats, see Tilman & Wedin [65]). Our analysis shows that height clustering (and not height *per se*) at middle-range latitudes is a fingerprint of a balance between energy invested in either potential growth or other mechanisms that may help plants overcome competitors. For instance, when competitors are close relatives in dense herbaceous communities, selection may favour the evolution of a low leaf height. In these situations, 'for short conspecific herbs to exclude competitors from a highly productive site, they must possess alternative mechanisms to overcome competition, such as root competition or allelochemics' [30]. More generally, we would argue that functional trade-offs tend to evolve in regions of higher primary productivity, where the relative role of biological interactions (competition, parasitism, herbivory) is expected to be higher.

Competitive hierarchies have been theoretically investigated [66,67] and empirically demonstrated in herbaceous plant communities at much smaller spatial scales [65,68,69]. Other hierarchies have been also investigated in tree communities [70]. In some of these studies, particular trade-offs have been shown to maintain plant diversity and limiting similarity, which involves that competitive dominance may also lead to trait overdispersion. However, these theoretical results arise as a consequence of a particular trade-off definition. We believe our theoretical models are more general [22] and, in their diverse formulations, invariably lead to the opposite pattern: trait clustering. Interestingly, the relevant role of competitive dominance driven by species trait hierarchies has been also reported at much smaller spatial scales for forest trees along an altitudinal gradient in the French Alps [19]. Moreover, a study of the assembly of forest communities across East Asia shows that a phylogenetic-based species similarity index tends to be smaller the higher the minimum temperature of the coldest month is [71]. Although traits are not generally related to competitive abilities, and they are diverse in their functionality and in their response to environmental stress, these studies, together with our results, suggest that trait clustering is generally likely to occur where conditions for plant growth are less restrictive. Our models indicate that the process underlying this pattern is competitive dominance rather than Darwin's competition-similarity hypothesis, although it is likely that community assembly for other taxa may be driven by other biotic or environmental filters. For instance, phytoplankton communities from estuarine ecosystems [72] are more consistent with Darwin's seminal hypothesis since they appear to be driven by limiting similarity creating clumpy species coexistence [73,74]. Competitive hierarchies are, of course, not hard-wired in nature. Intransitivities may still play a key role in maintaining diversity in some systems [75–77].

In Capitán *et al.* [22], we demonstrated how different coexistence versus competition curves can be collapsed into a single curve. Here we showed that model predictions were quantitatively consistent with the observed decaying behaviour of the probability of local coexistence as overall competition intensity increases. This general scaling behaviour is typical for stochastic community models in the presence of both symmetrical [31,32] and asymmetrical competition, as we showed in our previous publication [22]. Here we tested this pattern at large geographical scales. The scaling allowed us to give a rough estimate of $\hat{\rho}$, an average ratio of inter- versus intra-specific competition (see figure 3a). Our indirect method is only able to estimate an average $\hat{\rho}$ across ecoregions. This average estimate is a highly aggregated parameter calculated from the whole dataset, and therefore, characterizing

European herbaceous plant communities. Although we expect high variability in its value between ecoregions, in a given ecoregion, the ratio of inter- versus intra-competition is expected to be, on average, about 0.04. Whenever direct empirical estimates of the ratio of inter- versus intra-competition are obtained, a few similar species are typically studied using small-scale field experiments [78,79]. It is, therefore, unsurprising that empirical estimations of this parameter tend to be higher than ours [7], but see also Volkov *et al.* [80] and Wang *et al.* [81]. Being able to provide rough estimates of this parameter at regional scales is also a novel result from our analysis. Our results are in agreement with a recent study of trees across six forest biomes where the authors found that trait variation is mostly related to competitive imbalances tending to drive inferior competitors to extinction [20]. Further work is required to better relate the average ratio of inter- versus intra-specific competition, which stabilizes species coexistence, to plant traits, and analyse how this aggregated parameter changes at increasing spatial scales and across taxa.

In this paper, we have explored several predictions from theoretical models aimed at describing plant dynamics, which have been derived and carefully studied in [22]. In total, we have contrasted four model predictions against reported herbaceous plant diversity across Europe. Our theoretical models represent a strong oversimplification of real plant community dynamics. However, in spite of disregarding the true complexity of these communities, our theoretical approach is useful, not only because it can reproduce macro-ecological, observational patterns with a small number of meaningful aggregated variables, but also because it provides new quantitative or qualitative predictions than may lead to new both empirical and observational studies. We look forward to seeing our simple trait-driven theory of competitive dominance being falsified (or not) in other ecological contexts. We humbly believe our message should be discussed within the context of the full scientific community interested in biodiversity research. Finding a theoretically robust and ecologically meaningful rapprochement between theory and data at relevant scales remains a challenge for ecology, and we trust that our work will inspire new contributions in this direction.

Data accessibility. The LEDA Traitbase is an open Internet database, and data can be downloaded from https://uol.de/en/landeco/research/leda. The database of Atlas Florae Europaeae that supports the findings of this study is available from the Committee for Mapping the Flora of Europe and Societas Biologica Fennica Vanamo (https://www.luomus.fi/en/publishing-atlas-florae-europaeae). MODIS data for actual ET are also publicly available (http://www.ntsg.umt.edu/project/modis/mod17.php). Data and code for replicability of our results is available at the Dryad Digital Repository: https://doi.org/10.5061/dryad.wpzgmsbjq [82].

Authors' contributions. J.A.C. and D.A. designed the research; J.A.C. and S.C. performed the statistical analysis; J.A.C., S.C., A.O. and D.A. analysed data and results; A.O. prepared data files and contributed materials; J.A.C. and D.A. wrote the paper.

Competing interests. We declare we have no competing interests.

Funding. This work was funded by the Spanish 'Ministerio de Economía y Competitividad' under the projects CGL2012-39964 and CGL2015-69043-P (D.A. and J.A.C.), by the Spanish 'Ministerio de Ciencia, Innovación y Universidades' under the project PGC2018-096577-B-I00 (D.A. and J.A.C.), and the Ramón y Cajal Fellowship program (RYC-2010-06545, D.A.). J.A.C. acknowledges partial financial support from the Department of Applied Mathematics (Universidad Politécnica de Madrid). S.C. acknowledges financial support from Banco Santander through grant no. PR87/19-22582.

Acknowledgements. The authors thank Mercedes Pascual for her insightful comments, and are indebted to Rohan Arthur, Han Olff, Joaquín Hortal and Fernando Valladares for their constructive criticism on earlier versions of this manuscript.

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
