## [Peer Review File · Royal Society Open Science]

Review History

RSOS-201361.R0 (Original submission)

Review form: Reviewer 1

Is the manuscript scientifically sound in its present form?

No

Are the interpretations and conclusions justified by the results?

No

Is the language acceptable?

Yes

Do you have any ethical concerns with this paper?

No

Have you any concerns about statistical analyses in this paper?

Yes

Recommendation?

Major revision is needed (please make suggestions in comments)

Comments to the Author(s)

As noted in my emails to the editor, I reviewed this paper in February at Ecology Letters. Because the paper has not been substantively changed since that review, I am re-submitting the same review text that I provided at Ecology Letters.

Summary:

The study uses a competition model, in which competitive hierarchy is related to maximum plant height, to explain variation in the fraction of regional diversity which is found at smaller scales across a large number of sites in and around Europe. These results are used to support the hypothesis that joint impacts of temperature and water availability on species trait syndromes and competitive hierarchies are primary drivers of local diversity in the region.

General Comments:

First off, let me say that I think that this is wonderfully elegant, simple, and tractable model. I also applaud you for applying mechanistic insight about the underlying traits in your model to structure how species are assumed to interact – i.e. by focusing on asymmetrical effects of height differences, rather than just average trait distance, as is often done in other trait-based studies. Additionally, I think that the overall pattern that you identify – i.e. significant increases in trait-based clustering at mid-latitudes – is quite interesting and convincing. Taken together, I think that this paper demonstrates some very interesting patterns, and includes a plausible, semi-mechanistic explanation of why that pattern might exist.

Nevertheless, I have a number of major concerns about how the model is presented and interpreted. In particular, I think that there are a number of very strong assumptions in this model that are either not tested or well-supported, or for which the ramifications of breaking those assumptions are not explored. And, indeed, in many cases, I think there is quite a bit of evidence suggesting that these assumptions are not met in real-world systems (including many papers that you cite). Although you note some of these potential problems near the end of the discussion, as far as I can tell, there is not much included in this paper to demonstrate that your results are robust to these concerns, nor is it clear what kinds of biases might appear as a result of these confounding processes. In order for the paper to be more convincing to me, I think that it would require a major restructuring that more closely focused on identifying and quantitatively testing these assumptions.

1. Maximum height as the primary competitive indicator

First and foremost, I think that focusing on height as the primary (and only?) indicator of competitive hierarchy is not well supported by existing literature on herbaceous plant competition. In my detailed comments below, I include a number of specific citations that discuss other putative influential processes and traits that are potentially independent of height – e.g. herbivore resistance, mineral resource requirements, and eco-evolutionary history. Although I am not generally opposed to applying a simplified model with a small number of traits (and, as I said above, although I very much appreciate the simplicity of your model, and the mechanistic insight that it uses to model height-based interactions), I think that a lot more text and analyses would need to be devoted to justifying and testing this strong, central assumption.

Additionally, I worry that using “maximum height” taken from a general database of plant traits is potentially problematic. First, maximum values are strongly influenced by sample size – thus, plants that have been sampled many times (e.g. common species) will almost always have greater maximum height values than those that have only been sampled a few times (e.g. rare species).

Thus, I would generally suggest using a quantile-based metric (e.g. 95th percentile height), as this value tends to be a bit more robust to sample size.

Moreover, as you note in the text, height is an enormously plastic trait. At the very least, the fact that different species in the database are likely to have been measured under different conditions will add substantial observation error to the analyses. Moreover, it is very likely to also add bias. For example, in my own work with these databases, I have found that because weedy species tend to occur and be measured disproportionately in ruderal areas, they also tend to have significantly higher reported heights than non-weedy species – despite the fact that the relationship is typically reversed at the site level. Ideally, height data should be standardised based on differences in measurement conditions among sites – although this is rarely possible in practice. At the very least, I would strongly urge an analysis of some subset of the data where heights have been measured under standardised conditions, in order to demonstrate that your reported results remain unchanged.

2. Effects of the regional species pool

As far as I understand it, the regional species pool is taken directly from observations (i.e. it is not an emergent result of the model, but rather is an input that is subsequently used to model local diversity). At the very least, I think it would be worth discussing in a bit more detail what kind of limitations this might have for your results – e.g. much of the literature on evolutionary biogeography would argue that once the regional pool has been assembled, the local composition is more or less pre-ordained, and local interactions have very impact on outcomes. Again, you note some of these concerns briefly in the end of the discussion, but as far as I can tell, you do not include justifications that would explain why these concerns do not apply to your model. If possible, I would suggest trying to test the effects of the regional pool on your results. For example, to what extent do the clustering patterns that you identify arise as a general response to the climate gradients you are studying, regardless of the size and composition of the regional pool? Apologies if I am missing a part of the analysis that does this already – I know that the randomization routine that you apply does a good job of identifying significant trait clustering, but I'm not sure that this quite gets at the question that I am asking here.

3. Concepts of “ideal” growth conditions

I would suggest being a bit more careful about how you discuss “ideal” conditions for plants. It is true that many herbaceous plants grow “best” when water availability is high, and temperatures do not drop too low. But, these patterns are by no means unimodal, nor are they universal. For example, environments that are too wet can lead plants to drown if their roots are saturated, and can cause early mortality and fast turnover due to, e.g. fungal infections. Likewise, high night time temperatures can lead to increases in respiration rates, thereby reducing overall growth. A good source that discusses many of these relationships is Lambers & Oliveira (2020), or other earlier additions. I think it would be a good idea to discuss your results in light of these more nuanced effects of climate on plant growth, rather than focusing on the “optima” that you discuss currently.

4. Discussion of Chesson's theory

Lastly – a minor point, but much of the abstract and introduction focuses on Chesson's coexistence theory. But, I am not sure that your analyses draw very heavily on this literature or concept – rather, it tests a much or common hypothesis related to how functional traits and competitive forcing are related. As I discuss below, these concepts has been explored using Chesson's theory, but it is by no means a major component of it (his theory, rather, focuses on partitioning different kinds of impacts on species invasion rates when rare). Moreover, I think that the empirical evidence supporting the link between his theory and the functional trait literature are weak at best. I would suggest either removing this discussion from your manuscript, or making the link with Chesson's theoretical models clearer.

Specific Comments:

17-20: I am not so sure that there is strong evidence that functional trait clustering corresponds to fitness differences (or for that matter, to niche differences, or to coexistence in general). It is true that Kraft et al. (2015) demonstrate such a relationship in one system, but their model is also unable to explain realised local coexistence at their site – Although they attribute this to spatial heterogeneity, an equally good explanation is that their model simply didn't fit the data very well, and that the realised values for niche and fitness differences are therefore not very meaningful. I know of very little evidence for the correlation between trait clustering and Chesson's metrics outside of this paper (or other papers that directly use the same dataset and model) – and, several papers (e.g. Letten et al. 2017) seem to suggest that no such relationship should be expected at all. And, to the best of my understanding, the Mayfield & Levine paper is primarily a perspectives piece, and does not offer much empirical proof for the hypothesized relationship. Do you have any more general evidence to support this statement, and this general framework?

20-22: I agree with this statement – although I suspect many macroecologists would disagree. E.g. the Ricklefs (2008) paper you cite in the discussion rather energetically argues that large-scale diversity is driven by geography and evolutionary history, and that local species interactions are largely irrelevant. I think that this perspective might be worth discussing here in more detail.

31-32: “within the limits of structural stability”: What do you mean by this? I.e. does this refer to physical stability of the plant (e.g. tree vs. grass growth forms), or to “structural stability” in the mathematical sense?

35-43: How much evidence is there in these sites that competitive hierarchy is determined primarily via traits related to height, rather than, e.g. competition for soil mineral resources? Is there a strong reason to believe that height is a better indicator of competitive interactions than any other trait in these systems?

53-58: Based on your other papers, my understanding is that in this model, competitive interactions are highly asymmetrical and follow a strict competitive hierarchy. While I agree that this structure will generally lead to a decline in local coexistence with stronger average competitive interaction strength, I do not believe that this result must be general – e.g. if the strict competitive hierarchy is removed, or if a trade-off is present (e.g. if species with lower competitive hierarchy also tend to have higher carrying capacity). This isn't necessarily a problem, nor does it mean that your model isn't useful, but I would be careful to point out that the negative relationship between realised coexistence and average competitive intensity is largely a tautological result of the model structure, rather than a general relationship that must appear in all systems.

113-115: Can you explain this a bit more ecologically? I.e. is there a biological reason that the log differences are more meaningful than differences in linear space?

127-128: “Larger plants capture more resources”: Again, I don't think that this is necessarily true. For example, in a system that is nitrogen limited, smaller plants often have much higher tissue nitrogen concentrations, and ultimately take up more total nitrogen, than do larger plants (e.g. small herbs vs. tall grasses). Similarly, competitive ability for water can be inversely related to size, specifically because larger plants with larger leaves often lose more water to transpiration, and therefore require higher soil water concentrations to persist. In both of these cases, I think that the problem is exacerbated because you use height (e.g. rather than total above- + below-ground biomass) as your size index. Your argument is potentially a bit more correct for light competition, though again, it will depend on how biomass is distributed as a function of height.

In any case, I don't think that one can axiomatically use this statement to justify that evolution should favour taller plants as the primary mechanism of resource competitive ability.

151-155: Does the model consider the possibility of variable immigration rates, and variable carrying capacities, among species? If not, how would fitting the model to data with variables rates and capacities influence the parameter estimates?

161-167: A minor point, but it seems like this is a 2-tailed test? In this case, I think that the "classic" way of testing at $\alpha = 0.05$ would be to use the 2.5% and 97.5% quartiles as the reference states, such that the combined probability of the lower plus upper tails equals 5%. Apologies if I am missing something.

184-185; 194-198: Again, I am not sure that it is correct to state that evapotranspiration is known to be the primary indicator of environmental constraints on plant growth at large scales. Others have posited, e.g. herbivore-based controls (Borer et al. 2014), soil chemistry-based controls (Laliberte et al. 2014), or biogeographic/evolutionary controls (Ricklefs 2008; McKenna et al. 2009).

I think you are probably correct that a rough correlation between growth rate and evapotranspiration rates exists, and that this correlation is a good general indicator for growth rate, all else being equal. But, there is a huge amount of residual variance that isn't captured by these trends, which makes it difficult to label certain conditions as being "optimal" for plant growth. For example, even "optimal" radiation, water availability, and temperatures can lead to declines in growth – e.g. due to high covariance with herbivore or fungal abundance.

242-243: I'm not sure I agree with this statement – Given, as you say, that one of the defining aspects of a community is that species can interact, it seems important that they should be studied at scales within which interactions are possible. E.g. if a study fails to find significant evidence for species interactions at a 50x50 km² scale, that hardly means that species interactions are irrelevant to community structure – it just means that if there are significant effects, they occur at scales that are too small to be captured within the grain used by the study.

Just as a simple example: Both *Typha* and *Arrhenatherum* can easily exist in the same 50x50km² grid cell, but they will almost certainly never interact at local scales, simply because they cannot grow under the same conditions. In other words, co-occurrence at one scale cannot necessarily be taken as evidence of co-occurrence at smaller scales.

257-258: This seems to be the first section where you suggest that the correspondence between height and competitive ability that you assume may not hold – But, you don't discuss how breaking this assumption might alter your results. At the very least, I would suggest including a somewhat longer justification of why you think that your results demonstrate that height does indeed universally correspond with competitive hierarchy, and discuss what your results would have looked like had this assumption not been met.

274-276: I'm not totally sure that this is correct – e.g. the Tilman results that you cite generally suggest the existence of some kind of limiting similarity in competitive abilities, which would lead to disaggregation of traits.

Fig. 5: Apologies if I am missing something – but I think there is either a typo in the legend or in the figure? The legend says that panel (d) shows "Correlation between mean gross primary productivity (GPP) and mean annual ET", but the figure shows ET on both axes.

Spelling and Grammar:

238: “concept of ecological community” -> “concept of ecological communities”, or “ecological community concept”?

References:

Lambers, H, and Rafael S Oliveira. *Plant Physiological Ecology*. Cham: Springer, 2020.
 Letten, Andrew D., Po-Ju Ke, and Tadashi Fukami. “Linking Modern Coexistence Theory and Contemporary Niche Theory.” *Ecological Monographs* 87, no. 2 (May 2017): 161–77.
 Borer, Elizabeth T., Eric W. Seabloom, Daniel S. Gruner, W. Stanley Harpole, Helmut Hillebrand, Eric M. Lind, Peter B. Adler, et al. “Herbivores and Nutrients Control Grassland Plant Diversity via Light Limitation.” *Nature* 508, no. 7497 (April 2014): 517–20.
 Laliberté, E., G. Zemunik, and B. L. Turner. “Environmental Filtering Explains Variation in Plant Diversity along Resource Gradients.” *Science* 345, no. 6204 (September 26, 2014).

McKenna, Duane D., Andrea S. Sequeira, Adriana E. Marvaldi, and Brian D. Farrell. “Temporal Lags and Overlap in the Diversification of Weevils and Flowering Plants.” *Proceedings of the National Academy of Sciences* 106, no. 17 (April 28, 2009): 7083–88.

Decision letter (RSOS-201361.R0)

Dear Dr Capitan

The Editors assigned to your paper RSOS-201361 "A signal of competitive dominance in mid-latitude herbaceous plant communities" have now received comments from reviewers and would like you to revise the paper in accordance with the reviewer comments and any comments from the Editors. Please note this decision does not guarantee eventual acceptance.

Please submit your revised manuscript and required files (see below) no later than 21 days from today's (ie 06-Jan-2021) date. Note: the ScholarOne system will 'lock' if submission of the revision is attempted 21 or more days after the deadline. If you do not think you will be able to meet this deadline please contact the editorial office immediately.

Please note article processing charges apply to papers accepted for publication in Royal Society Open Science (<https://royalsocietypublishing.org/rsos/charges>). Charges will also apply to papers transferred to the journal from other Royal Society Publishing journals, as well as papers submitted as part of our collaboration with the Royal Society of Chemistry

(<https://royalsocietypublishing.org/rsos/chemistry>). Fee waivers are available but must be requested when you submit your revision (<https://royalsocietypublishing.org/rsos/waivers>).

on behalf of Professor Brian Reid (Associate Editor) and Pete Smith (Subject Editor)
openscience@royalsociety.org

Reviewer comments to Author:

Reviewer: 1

Comments to the Author(s)

As noted in my emails to the editor, I reviewed this paper in February at Ecology Letters. Because the paper has not been substantively changed since that review, I am re-submitting the same review text that I provided at Ecology Letters.

Summary:

The study uses a competition model, in which competitive hierarchy is related to maximum plant height, to explain variation in the fraction of regional diversity which is found at smaller scales across a large number of sites in and around Europe. These results are used to support the hypothesis that joint impacts of temperature and water availability on species trait syndromes and competitive hierarchies are primary drivers of local diversity in the region.

General Comments:

First off, let me say that I think that this is wonderfully elegant, simple, and tractable model. I also applaud you for applying mechanistic insight about the underlying traits in your model to structure how species are assumed to interact – i.e. by focusing on asymmetrical effects of height differences, rather than just average trait distance, as is often done in other trait-based studies. Additionally, I think that the overall pattern that you identify – i.e. significant increases in trait-based clustering at mid-latitudes – is quite interesting and convincing. Taken together, I think that this paper demonstrates some very interesting patterns, and includes a plausible, semi-mechanistic explanation of why that pattern might exist.

Nevertheless, I have a number of major concerns about how the model is presented and interpreted. In particular, I think that there are a number of very strong assumptions in this model that are either not tested or well-supported, or for which the ramifications of breaking those assumptions are not explored. And, indeed, in many cases, I think there is quite a bit of evidence suggesting that these assumptions are not met in real-world systems (including many papers that you cite). Although you note some of these potential problems near the end of the discussion, as far as I can tell, there is not much included in this paper to demonstrate that your results are robust to these concerns, nor is it clear what kinds of biases might appear as a result of these confounding processes. In order for the paper to be more convincing to me, I think that it would require a major restructuring that more closely focused on identifying and quantitatively testing these assumptions.

1. Maximum height as the primary competitive indicator

First and foremost, I think that focusing on height as the primary (and only?) indicator of competitive hierarchy is not well supported by existing literature on herbaceous plant competition. In my detailed comments below, I include a number of specific citations that discuss other putative influential processes and traits that are potentially independent of height – e.g. herbivore resistance, mineral resource requirements, and eco-evolutionary history. Although I am not generally opposed to applying a simplified model with a small number of traits (and, as I said above, although I very much appreciate the simplicity of your model, and the mechanistic insight that it uses to model height-based interactions), I think that a lot more text and analyses would need to be devoted to justifying and testing this strong, central assumption.

Additionally, I worry that using “maximum height” taken from a general database of plant traits is potentially problematic. First, maximum values are strongly influenced by sample size – thus, plants that have been sampled many times (e.g. common species) will almost always have greater maximum height values than those that have only been sampled a few times (e.g. rare species). Thus, I would generally suggest using a quantile-based metric (e.g. 95th percentile height), as this value tends to be a bit more robust to sample size.

Moreover, as you note in the text, height is an enormously plastic trait. At the very least, the fact that different species in the database are likely to have been measured under different conditions will add substantial observation error to the analyses. Moreover, it is very likely to also add bias. For example, in my own work with these databases, I have found that because weedy species tend to occur and be measured disproportionately in ruderal areas, they also tend to have significantly higher reported heights than non-weedy species – despite the fact that the relationship is typically reversed at the site level. Ideally, height data should be standardised based on differences in measurement conditions among sites – although this is rarely possible in practice. At the very least, I would strongly urge an analysis of some subset of the data where heights have been measured under standardised conditions, in order to demonstrate that your reported results remain unchanged.

2. Effects of the regional species pool

As far as I understand it, the regional species pool is taken directly from observations (i.e. it is not an emergent result of the model, but rather is an input that is subsequently used to model local diversity). At the very least, I think it would be worth discussing in a bit more detail what kind of limitations this might have for your results – e.g. much of the literature on evolutionary biogeography would argue that once the regional pool has been assembled, the local composition is more or less pre-ordained, and local interactions have very impact on outcomes. Again, you note some of these concerns briefly in the end of the discussion, but as far as I can tell, you do not include justifications that would explain why these concerns do not apply to your model. If possible, I would suggest trying to test the effects of the regional pool on your results. For example, to what extent do the clustering patterns that you identify arise as a general response to the climate gradients you are studying, regardless of the size and composition of the regional pool? Apologies if I am missing a part of the analysis that does this already – I know that the randomization routine that you apply does a good job of identifying significant trait clustering, but I’m not sure that this quite gets at the question that I am asking here.

3. Concepts of “ideal” growth conditions

I would suggest being a bit more careful about how you discuss “ideal” conditions for plants. It is true that many herbaceous plants grow “best” when water availability is high, and temperatures do not drop too low. But, these patterns are by no means unimodal, nor are they universal. For example, environments that are too wet can lead plants to drown if their roots are saturated, and can cause early mortality and fast turnover due to, e.g. fungal infections. Likewise, high night time temperatures can lead to increases in respiration rates, thereby reducing overall growth. A good source that discusses many of these relationships is Lambers & Oliveira (2020), or other

earlier additions. I think it would be a good idea to discuss your results in light of these more nuanced effects of climate on plant growth, rather than focusing on the “optima” that you discuss currently.

4. Discussion of Chesson’s theory

Lastly – a minor point, but much of the abstract and introduction focuses on Chesson’s coexistence theory. But, I am not sure that your analyses draw very heavily on this literature or concept – rather, it tests a much or common hypothesis related to how functional traits and competitive forcing are related. As I discuss below, these concepts has been explored using Chesson’s theory, but it is by no means a major component of it (his theory, rather, focuses on partitioning different kinds of impacts on species invasion rates when rare). Moreover, I think that the empirical evidence supporting the link between his theory and the functional trait literature are weak at best. I would suggest either removing this discussion from your manuscript, or making the link with Chesson’s theoretical models clearer.

Specific Comments:

17-20: I am not so sure that there is strong evidence that functional trait clustering corresponds to fitness differences (or for that matter, to niche differences, or to coexistence in general). It is true that Kraft et al. (2015) demonstrate such a relationship in one system, but their model is also unable to explain realised local coexistence at their site – Although they attribute this to spatial heterogeneity, an equally good explanation is that their model simply didn’t fit the data very well, and that that the realised values for niche and fitness differences are therefore not very meaningful. I know of very little evidence for the correlation between trait clustering and Chesson’s metrics outside of this paper (or other papers that directly use the same dataset and model) – and, several papers (e.g. Letten et al. 2017) seem to suggest that no such relationship should be expected at all. And, to the best of my understanding, the Mayfield & Levine paper is primarily a perspectives piece, and does not offer much empirical proof for the hypothesized relationship. Do you have any more general evidence to support this statement, and this general framework?

20-22: I agree with this statement – although I suspect many macroecologists would disagree. E.g. the Ricklefs (2008) paper you cite in the discussion rather energetically argues that large-scale diversity is driven by geography and evolutionary history, and that local species interactions are largely irrelevant. I think that this perspective might be worth discussing here in more detail.

31-32: “within the limits of structural stability”: What do you mean by this? I.e. does this refer to physical stability of the plant (e.g. tree vs. grass growth forms), or to “structural stability” in the mathematical sense?

35-43: How much evidence is there in these sites that competitive hierarchy is determined primarily via traits related to height, rather than, e.g. competition for soil mineral resources? Is there a strong reason to believe that height is a better indicator of competitive interactions than any other trait in these systems?

53-58: Based on your other papers, my understanding is that in this model, competitive interactions are highly asymmetrical and follow a strict competitive hierarchy. While I agree that this structure will generally lead to a decline in local coexistence with stronger average competitive interaction strength, I do not believe that this result must be general – e.g. if the strict competitive hierarchy is removed, or if a trade-off is present (e.g. if species with lower competitive hierarchy also tend to have higher carrying capacity). This isn’t necessarily a problem, nor does it mean that your model isn’t useful, but I would be careful to point out that the negative relationship between realised coexistence and average competitive intensity is

largely a tautological result of the model structure, rather than a general relationship that must appear in all systems.

113-115: Can you explain this a bit more ecologically? I.e. is there a biological reason that the log differences are more meaningful than differences in linear space?

127-128: “Larger plants capture more resources”: Again, I don’t think that this is necessarily true. For example, in a system that is nitrogen limited, smaller plants often have much higher tissue nitrogen concentrations, and ultimate take up more total nitrogen, than do larger plants (e.g. small herbs vs. tall grasses). Similarly, competitive ability for water can be inversely related to size, specifically because larger plants with larger leaves often lose more water to transpiration, and therefore require higher soil water concentrations to persist. In both of these cases, I think that the problem is exacerbated because you use height (e.g. rather than total above- + below-ground biomass) as your size index. Your argument is potentially a bit more correct for light competition, though again, it will depend on how biomass is distributed as a function of height. In any case, I don’t think that one can axiomatically use this statement to justify that evolution should favour taller plants as the primary mechanism of resource competitive ability.

151-155: Does the model consider the possibility of variable immigration rates, and variable carrying capacities, among species? If not, how would fitting the model to data with variables rates and capacities influence the parameter estimates?

161-167: A minor point, but it seems like this is a 2-tailed test? In this case, I think that the “classic” way of testing at $\alpha = 0.05$ would be to use the 2.5% and 97.5% quartiles as the reference states, such that the combined probability of the lower plus upper tails equals 5%. Apologies if I am missing something.

184-185; 194-198: Again, I am not sure that it is correct to state that evapotranspiration is known to be the primary indicator of environmental constraints on plant growth at large scales. Others have posited, e.g. herbivore-based controls (Borer et al. 2014), soil chemistry-based controls (Laliberte et al. 2014), or biogeographic/evolutionary controls (Ricklefs 2008; McKenna et al. 2009).

I think you are probably correct that a rough correlation between growth rate and evapotranspiration rates exists, and that this correlation is a good general indicator for growth rate, all else being equal. But, there is a huge amount of residual variance that isn’t captured by these trends, which makes it difficult to label certain conditions as being “optimal” for plant growth. For example, even “optimal” radiation, water availability, and temperatures can lead to declines in growth – e.g. due to high covariance with herbivore or fungal abundance.

242-243: I’m not sure I agree with this statement – Given, as you say, that one of the defining aspects of a community is that species can interact, it seems important that they should be studied at scales within which interactions are possible. E.g. if a study fails to find significant evidence for species interactions at a 50x50 km² scale, that hardly means that species interactions are irrelevant to community structure – it just means that if there are significant effects, they occur at scales that are too small to be captured within the grain used by the study.

Just as a simple example: Both *Typha* and *Arrhenatherum* can easily exist in the same 50x50km² grid cell, but they will almost certainly never interact at local scales, simply because they cannot grow under the same conditions. In other words, co-occurrence at one scale cannot necessarily be taken as evidence of co-occurrence at smaller scales.

257-258: This seems to be the first section where you suggest that the correspondence between height and competitive ability that you assume may not hold – But, you don't discuss how breaking this assumption might alter your results. At the very least, I would suggest including a somewhat longer justification of why you think that your results demonstrate that height does indeed universally correspond with competitive hierarchy, and discuss what your results would have looked like had this assumption not been met.

274-276: I'm not totally sure that this is correct – e.g. the Tilman results that you cite generally suggest the existence of some kind of limiting similarity in competitive abilities, which would lead to disaggregation of traits.

Fig. 5: Apologies if I am missing something – but I think there is either a typo in the legend or in the figure? The legend says that panel (d) shows “Correlation between mean gross primary productivity (GPP) and mean annual ET”, but the figure shows ET on both axes.

Spelling and Grammar:

238: “concept of ecological community” -> “concept of ecological communities”, or “ecological community concept”?

References:

Lambers, H, and Rafael S Oliveira. *Plant Physiological Ecology*. Cham: Springer, 2020.

Letten, Andrew D., Po-Ju Ke, and Tadashi Fukami. “Linking Modern Coexistence Theory and Contemporary Niche Theory.” *Ecological Monographs* 87, no. 2 (May 2017): 161–77.

Borer, Elizabeth T., Eric W. Seabloom, Daniel S. Gruner, W. Stanley Harpole, Helmut Hillebrand, Eric M. Lind, Peter B. Adler, et al. “Herbivores and Nutrients Control Grassland Plant Diversity via Light Limitation.” *Nature* 508, no. 7497 (April 2014): 517–20.

Laliberte, E., G. Zemunik, and B. L. Turner. “Environmental Filtering Explains Variation in Plant Diversity along Resource Gradients.” *Science* 345, no. 6204 (September 26, 2014).

McKenna, Duane D., Andrea S. Sequeira, Adriana E. Marvaldi, and Brian D. Farrell. “Temporal Lags and Overlap in the Diversification of Weevils and Flowering Plants.” *Proceedings of the National Academy of Sciences* 106, no. 17 (April 28, 2009): 7083–88.

===PREPARING YOUR MANUSCRIPT===

While not essential, it will speed up the preparation of your manuscript proof if accepted if you format your references/bibliography in Vancouver style (please see

<https://royalsociety.org/journals/authors/author-guidelines/#formatting>). You should include DOIs for as many of the references as possible.

===PREPARING YOUR REVISION IN SCHOLARONE===

Author's Response to Decision Letter for (RSOS-201361.R0)

See Appendix A.

RSOS-201361.R1 (Revision)

Review form: Reviewer 2 (Ilya Maclean)

Is the manuscript scientifically sound in its present form?

No

Are the interpretations and conclusions justified by the results?

No

Is the language acceptable?

Yes

Do you have any ethical concerns with this paper?

No

Have you any concerns about statistical analyses in this paper?

No

Recommendation?

Major revision is needed (please make suggestions in comments)

Comments to the Author(s)

The authors present a model of competitive dominance that predicts some interesting macroecological patterns, which are then tested against empirical data. The model is elegant in its simplicity and quite compelling. In essence the authors assume vegetation height to be the key indicator of competitive strength and show that, at macro scales, this is related to light and evapotranspiration and find that plant height clustering is most evident in mid-latitude ecoregions, where they assume conditions for growth (reflected in actual evapotranspiration rates

and gross primary productivities) are optimal. In effect they build a simple model and show that the pattern predicted by the model is the same as that observed. This is potentially really neat, but I don't think this is really the same as truly validating the model – it just so happens that the patterns are broadly similar, which could be coincidental, or could be that the model is good. The pattern could be explained by other things: e.g. abiotic rather than biotic controls or cause rather than effect. The authors assume mean annual evapotranspiration is a reliable measure of environmental constraints on plant growth. However, while partially related to leaf temperature (and hence net radiation and climate), evapotranspiration is strongly controlled by stomatal conductance (and hence photosynthetic rates) and foliage density (and hence canopy height). It could instead be argued that evapotranspiration rates are predictable from vegetation growth rather than the other way around.

Beyond that it is quite hard to get a sense of this paper. This is partly as it is really hard to follow as most of theoretical grounding presented in another paper, the methods themselves are not detailed enough and don't list all of the datasets used, and the results are a confusing blend of theory, method and actual results. Other potential limitations are that (i) vegetation height is assumed the key (only) indicator of competitive strength, when quite clearly this may not be the case and (ii) it is not clear whether the latitudinal patterns of potential light and water availability presented in Fig 1 come from. Is this PAR or some other measure of light availability, and is it constrained to seasons of plant growth (potentially revealing much higher light levels in the Arctic owing to 24 daylight)?

Most of the above could be handled with a convincing rebuttal, some more nuanced presentation of concepts and with a significant restructuring of the manuscript. I'd suggest restructuring the middle bit as follows: (1) outline the theory, (2) detail patterns predicted by the theory, (3) list the data obtained to test the theory (including all the environmental data) and (4) present the results (i.e. extent to which data support theory). (2) and (3) could potentially swap order, but really needs to know about the theory before finding out about the data.

In summary, this could be quite a neat paper, but it could also be quite flawed, and I would really need to see a more sensibly structured manuscript before fully passing judgement.

Decision letter (RSOS-201361.R1)

Dear Dr Capitan

The Editors assigned to your paper RSOS-201361.R1 "A signal of competitive dominance in mid-latitude herbaceous plant communities" have now received comments from reviewers and would like you to revise the paper in accordance with the reviewer comments and any comments from the Editors. Please note this decision does not guarantee eventual acceptance.

Please submit your revised manuscript and required files (see below) no later than 21 days from today's (ie 14-Jun-2021) date. Note: the ScholarOne system will 'lock' if submission of the revision is attempted 21 or more days after the deadline. If you do not think you will be able to meet this deadline please contact the editorial office immediately.

on behalf of Professor Pete Smith (Subject Editor)
openscience@royalsociety.org

Associate Editor Comments to Author:

Thank you for your further patience with the review of your work - it has unfortunately been very difficult to find referees for the paper: indeed, the original reviewer of your paper was unavailable to assess the revision, and we've struggled until recently to find a replacement. The new reviewer has provided thoughtful commentary on your work, and - though we don't generally permit multiple rounds of revision - we would like you to revise your paper. This will, however, be the last chance you receive to get the paper to a publishable standard. If you can persuade the reviewer that the promise they see in your work has been realised, great! If they continue to express similar concerns after revision, we may not be able to consider the paper further. With this in mind, please be careful in preparing your revision to clearly identify the changes you make (both in a tracked changes version of the paper and a point-by-point rebuttal). Good luck and we'll look forward to receiving the review in the near future.

Reviewer comments to Author:

Reviewer: 2

Comments to the Author(s)

The authors present a model of competitive dominance that predicts some interesting macroecological patterns, which are then tested against empirical data. The model is elegant in its simplicity and quite compelling. In essence the authors assume vegetation height to be the key indicator of competitive strength and show that, at macro scales, this is related to light and evapotranspiration and find that plant height clustering is most evident in mid-latitude ecoregions, where they assume conditions for growth (reflected in actual evapotranspiration rates and gross primary productivities) are optimal. In effect they build a simple model and show that

the pattern predicted by the model is the same as that observed. This is potentially really neat, but I don't think this is really the same as truly validating the model – it just so happens that the patterns are broadly similar, which could be coincidental, or could be that the model is good. The pattern could be explained by other things: e.g. abiotic rather than biotic controls or cause rather than effect. The authors assume mean annual evapotranspiration is a reliable measure of environmental constraints on plant growth. However, while partially related to leaf temperature (and hence net radiation and climate), evapotranspiration is strongly controlled by stomatal conductance (and hence photosynthetic rates) and foliage density (and hence canopy height). It could instead be argued that evapotranspiration rates are predictable from vegetation growth rather than the other way around.

Beyond that it is quite hard to get a sense of this paper. This is partly as it is really hard to follow as most of theoretical grounding presented in another paper, the methods themselves are not detailed enough and don't list all of the datasets used, and the results are a confusing blend of theory, method and actual results. Other potential limitations are that (i) vegetation height is assumed the key (only) indicator of competitive strength, when quite clearly this may not be the case and (ii) it is not clear whether the latitudinal patterns of potential light and water availability presented in Fig 1 come from. Is this PAR or some other measure of light availability, and is it constrained to seasons of plant growth (potentially revealing much higher light levels in the Arctic owing to 24 daylight)?

Most of the above could be handled with a convincing rebuttal, some more nuanced presentation of concepts and with a significant restructuring of the manuscript. I'd suggest restructuring the middle bit as follows: (1) outline the theory, (2) detail patterns predicted by the theory, (3) list the data obtained to test the theory (including all the environmental data) and (4) present the results (i.e. extent to which data support theory). (2) and (3) could potentially swap order, but really needs to know about the theory before finding out about the data.

In summary, this could be quite a neat paper, but it could also be quite flawed, and I would really need to see a more sensibly structured manuscript before fully passing judgement.

===PREPARING YOUR MANUSCRIPT===

===PREPARING YOUR REVISION IN SCHOLARONE===

Author's Response to Decision Letter for (RSOS-201361.R1)

See Appendix B.

Decision letter (RSOS-201361.R2)

Dear Dr Capitan,

It is a pleasure to accept your manuscript entitled "A signal of competitive dominance in mid-latitude herbaceous plant communities" in its current form for publication in Royal Society Open Science. The comments from the Editors are included at the foot of this letter.

Please ensure that you send to the editorial office an editable version of your accepted manuscript, and individual files for each figure and table included in your manuscript. You can send these in a zip folder if more convenient. Failure to provide these files may delay the processing of your proof.

Please see the Royal Society Publishing guidance on how you may share your accepted author manuscript at <https://royalsociety.org/journals/ethics-policies/media-embargo/>. After publication, some additional ways to effectively promote your article can also be found here

<https://royalsociety.org/blog/2020/07/promoting-your-latest-paper-and-tracking-your-results/>.

on behalf of Professor Leslie Brown (Associate Editor) and Pete Smith (Subject Editor)
openscience@royalsociety.org

Associate Editor Comments to Author (Professor Leslie Brown):

Dear authors, thank you for the detailed comments/explanations to each of the reviewers comments/suggestions and the positive way in which you addressed it. I am of the opinion that the changes affected improves the manuscript and its value considerably. I have made my recommendation and sent it to the editor for final decision. All of the best with your research and this interesting manuscript.

Follow Royal Society Publishing on Twitter: [@RSocPublishing](https://twitter.com/RSocPublishing)

Appendix A

Abstract

Understanding the main determinants of species coexistence across space and time is a central question in ecology. However, ecologists still know little about the scales and conditions at which biotic interactions matter and how these interact with the environment to structure species assemblages. Here we use recent theory developments to analyze plant distribution and trait data across Europe and find that plant height clustering is related to both evapotranspiration and gross primary productivity. This clustering is a signal of interspecies competition between plants, which is most evident in mid-latitude ecoregions, where conditions for growth (reflected in actual evapotranspiration rates and gross primary productivities) are optimal. Away from this optimum, climate severity likely overrides the effect of competition, or other interactions become increasingly important. Our approach bridges the gap between modern coexistence theory and large-scale species distribution data analysis.

Keywords: Ecological community dynamics Plant diversity Species coexistence Biogeographic patterns Null hypotheses testing Stochastic Markov processes in continuous time.

Introduction

Modern coexistence theory (Chesson, 2000, HilleRisLambers *et al.*, 2011, Mayfield & Levine, 2010) is based
on species difference and their interplay to determine effective competitive (biotic) interactions among species in
natural communities. The balance between stabilizing trait differences and species dominance among competi-
tors is crucial to understand species coexistence under this framework. In communities driven by fitness differ-
ences, species turn out to be clustered around similar trait values selected through competitive dominance. Trait
clustering may arise through two radically different mechanisms. One possible explanation is that it would arise
through independent adaptation of non-interacting species to the same environmental conditions. The alternative
explanation would say that competitive interactions leading to fitness equalization end up producing more similar
species, with, therefore, more similar traits. Therefore, trait clustering may be interpreted as a fingerprint of com-
petition even in the absence of environmental filtering (Kraft *et al.*, 2015, Mayfield & Levine, 2010). Although the
role of local interactions at determining large-scale diversity patterns is still controversial Ricklefs (2008). Com-
munity ecology, however, ~~still-needs-lacks~~ a comprehensive theoretical framework able to ~~describe quantitatively~~
explore quantitatively to what extent the role of biotic, species-to-species interactions ~~that-are-is~~ relevant to deter-
mine species composition and diversity across large spatial scales. Empirical studies, while they may be able to
independently assess environmental stress and species competitive abilities, are often limited to small community
sizes (Violle *et al.*, 2011) or restricted to single habitats (Kunstler *et al.*, 2012). Very few studies have explored
the idea of competition as a driver of community assembly across biogeographic regions (Kunstler *et al.*, 2016,
Swenson *et al.*, 2012). Here we attempted a continent-wide macro-ecological study of species assemblage patterns
based on theoretical predictions from modern coexistence theory (Capitán *et al.*, 2020, Chesson, 2000, Mayfield
& Levine, 2010) at large geographical scales.

Light and water availability (Fig. 1) impose significant limitations on gross primary productivity which is
reflected in actual evapotranspiration rates (Garbulsky *et al.*, 2010). These two resources vary at regional scales,
placing strong, sometimes opposing constraints on how tall a plant can grow ~~within the limits of structural stability~~.
Plant height is a fundamental trait that reflects the ability of the individual to optimize its own growth within its
local biotic environment and regional physical constraints (see Falster & Westoby (2003), Holmgren *et al.* (1997)
and references therein). How plant height adapts to these opposing constraints has been studied in trees (King,
1990, Law *et al.*, 1997, Midgley, 2003) and herbaceous plants (Givnish, 1995, 1982). Here we analyzed presence-
absence matrices of floral herbaceous taxa across different European ecoregions to determine if competitive ability
(reflected in maximum stem height) could help explain assemblage patterns at local scales across gradients of
relevant environmental factors such as evapotranspiration. We examined how well observed plant assemblages at
macro-ecological scales match theoretical predictions generated by a synthetic, stochastic framework of commu-
nity assembly (Capitán *et al.*, 2015, 2017, Haegeman & Loreau, 2011, McKane *et al.*, 2000), which we described
in full detail in Capitán *et al.* (2020). By assuming that competition between hetero-specifics is driven by signed

[revised manuscript text omitted]

**Materials and methods**

Plant community data were drawn from Atlas Florae Europaeae (Jalas & Suominen, 1964–1999). The distribution
of flora is geographically described using equally-sized grid cells ($\sim 50 \times 50$ km) based on the Universal Transverse
Mercator projection and the Military Grid Reference System, see Fig. 2. Each cell was assigned to a dominant
habitat type based on the WWF Biomes of the World classification (Olson *et al.*, 2001), which defines different
ecoregions, i.e., geographically distinct assemblages of species subject to similar environmental conditions. We
consider each cell in an ecoregion to represent a species aggregation.

Each herbaceous species in an ecoregion was characterized by its maximum stem height H , an eco-morphological
trait that relates to several critical functional strategies among plants (Díaz *et al.*, 2015). It represents an optimal
trade-off between the gains of accessing light (King, 1990, Law *et al.*, 1997), water and nutrient transport from
soil (Midgley, 2003, Ryan & Yoder, 1997), and additional constraints posed by the local biotic environment of

each individual plant, such as competition, facilitation, or herbivory.

Mean height values were obtained from the LEDA database (Kleyer *et al.*, 2008) for as many species as there
were available in the database. Missing values were taken from (Ordóñez *et al.*, 2010) or inferred using a MICE
(Multivariate Imputation by Chained Equations) approach (Buuren & Groothuis-Oudshoorn, 2011) together with
a predictive mean matching algorithm based on other available traits (leaf and seed traits), genus, and growth
forms as predictors. Based on plant growth forms, 2610 herbaceous species (aquatic, herbs, or graminoid) were
considered in this work.

Maximum stem height values spanned several orders of magnitude, so we used a log-transformed variable
($h = \log H$) to measure species differences (using non-transformed heights yielded comparable results, here not
shown). The values of h were standardized within ecoregions as $t = (h - h_{\min}) / (h_{\max} - h_{\min})$ so that $0 \leq t \leq 1$.

Results

For all the species reported in an ecoregion, we formed an empirical competition matrix with pairwise ρ_{ij} signed
height differences $\rho_{ij} = \hat{\rho}(t_j - t_i)$, where t_i are height values standardized across ecoregions and sorted in increas-
ing order. The advantage of having these values represent trait differences between pairs of species is that any trend
in competitive strengths can be immediately translated into patterns of functional clustering or overdispersion. As
in Capitán *et al.* (2020), we calculated the average competitive strength as $\langle \rho \rangle = \frac{2}{S(S-1)} \sum_{i=1}^S \sum_{j=i+1}^S |\rho_{ij}|$, S
standing for ecoregion richness.

In an ecoregion with richness S , a number $s_k \leq S$ of species will form a species assemblage at cell k . The
coexistence probability was calculated from data as the average fraction of species that survive per cell,

$$p_c = \frac{\langle s \rangle}{S} = \frac{1}{SN_C} \sum_{k=1}^{N_C} s_k, \quad (2)$$

with N_C representing the number of cells in the ecoregion. This quantity, together with the distribution of trait
differences in cells, was used to compare model predictions with real data.

~~Larger~~ If larger plants capture more resources.—~~Therefore~~, evolution should favor investment in potential
growth (maximum height) as a competitive mechanism. However, investment in alternative mechanisms, such as
allelopathy, may help smaller plants stave off competitors, reducing local heterospecific plant cover and giving
them a competitive advantage over potentially taller plant species. As a consequence, the maximum species stem
height can be regarded as the outcome of an evolutionary game (Givnish, 1982) that balances opposing constraints,
both physical (Craine & Dybzinski, 2013, Falster & Westoby, 2003) and biotic (King, 1990, Law *et al.*, 1997). To
explore these opposing constraints, we analyzed plant data in the light of the two community assembly models.
The first one is a spatially-implicit model of Lotka-Volterra type, and the second one is a straightforward spatially-
explicit extension including height-driven competition and allelopathic effects. Both have been carefully defined

and studied in Capitán *et al.* (2020).

**Two predictions from the implicit model tested against data**

The collapse of curves predicted by Eq. (1) helps eliminate the variability in S , so that empirical coexistence
probabilities, which arise from different ecoregion sizes, can be fitted together (Fig. 3). Confirming the first
prediction of the spatially-implicit model, we found a significant correlation between the probability of coexistence
and the scaled competitive overlap based on empirical data (Fig. 3), indicating that a model driven solely by
dominant competitive interactions reliably predicts the average richness of plant communities across ecoregions.
In addition, this theoretical prediction allowed an indirect estimation of the relative importance $\hat{\rho}$ of average inter-
vs. intraspecific effects: the average ratio of inter- to intraspecific competition strength is about 5% (see Supporting
Information, section A for details on the estimation procedure).

As a second prediction, the implicit model implies high levels of trait clustering for low immigration rates and
high carrying capacity values. Importantly, this parameter regime precisely emerges from the data. In Capitán
*et al.* (2020) we derived a deterministic prediction for the exponent, $\gamma = 1$, which does not match the one obtained
from data ($\gamma = 0.61$). As shown in that paper, it is a non-zero (but small) value of the immigration rate that
determines the power-law exponent γ being lower than 1. Indeed, for a realistic fit in Fig. 3, the exponent of the
empirical power law is obtained for $\mu/\alpha \sim 0.1$ individuals per generation. Since plant communities operate in
a low-immigration regime, the non-dimensional immigration rate $\lambda = \mu/(\alpha K)$ must satisfy $\lambda = 0.1/K \ll 1$,
hence the carrying capacity must be large. In a regime of low immigration rate and high carrying capacity, which
best fits empirical coexistence probabilities, the implicit model predicts a significant degree of species clustering
[see Fig. 3 in Capitán *et al.* (2020)].

Following Triadó-Margarit *et al.* (2019), our randomization tests applied to empirical communities were based
on the average competitive strength observed in a cell C formed by s species,

$$\langle \rho \rangle_C = \frac{2}{s(s-1)} \sum_{i=1}^s \sum_{j=i+1}^s |\rho_{ij}^C|, \quad (3)$$

where (ρ_{ij}^C) is the submatrix of the ecoregion competition matrix restricted to the species present in the cell.
Compared to ecoregion samples, the lower (higher) the empirical community average $\langle \rho \rangle_C$ is, the higher (lower)
is the degree of species clustering in the cell. For each cell we calculated the probability $p = \Pr(\langle \rho \rangle_Q \leq \langle \rho \rangle_C)$
that the the competition average $\langle \rho \rangle_Q$ randomly-sampled from the pool is smaller than the empirical average. At
a 5% significance level, if $p > 0.95$ the empirical competition average is significantly larger than the average
measured for random pool samples, which implies that average trait differences in realized communities are larger
than would be expected at random. On the other hand, if $p < 0.05$, observed trait differences are significantly
smaller than would be expected at random. Therefore, if $p > 0.95$, the community exhibits ‘significant trait
overdispersion’, whereas if $p < 0.05$, there is evidence for ‘significant trait clustering’ in the observed species

assemblage.

Testing the second prediction against empirical observations yields a mixed picture. We calculated p -values for
randomization tests applied to every cell in each ecoregion, which represent the empirical distribution of p -values
(Fig. 4). At the parameter values that make plant data consistent with the first prediction, the spatially-implicit
model predicts significant trait clustering. We observe that some ecoregions are consistent with this theoretical
expectation. However, other ecoregions clearly do not comply with this prediction. In addition, no ecoregion is
consistent with trait overdispersion (Fig. 4). Selecting species in randomization tests according to species dispersal
abilities portrays the same picture (results not shown).

**Ecoregion clustering and actual evapotranspiration rates**

In order to better quantify the propensity of an ecoregion to exhibit clustering in maximum stem height, we defined
a clustering index q for an ecoregion as the fraction of its cells that lie within the 5% range of significant clustering
(randomization tests yield p -values smaller than 0.05 for those cells). An ecoregion for which significant clustering
is found in most of its cells will tend to score high in the q index. We examined how the clustering index varied
across the continent in terms of the geographical location of ecoregion centroids as well as with actual evapotran-
spiration (Fig. 5). Evapotranspiration maps were obtained from data estimated through remote sensing (Mu *et al.*,
2011).

Water availability acts as a factor limiting plant growth at geographical scales (Fig. 1a), and correlates with
gross primary productivity (Garbulsky *et al.*, 2010), see Fig. 5d. Therefore, for a given region, mean annual evap-
otranspiration is a reliable measure of environmental constraints on plant growth (Garbulsky *et al.*, 2010). Panels a
and b of Fig. 5 show a clear latitudinal trend: there is an intermediate range of ecoregion latitudes where both clus-
tering indices and evapotranspiration are large, indicating that evapotranspiration measures can robustly predict
clustering indices (Fig. 5c). The same pattern can also be seen in the relation between mean relative height differ-
ences and actual evapotranspiration across individual grid cells. The intensity of the clustering pattern increases
with actual evapotranspiration rates across Europe, not only at the ecoregional level (Fig. 5c), but also at the lower
spatial scale of grid cells (see Fig. C1, Supporting Information). More importantly, since evapotranspiration is
a powerful proxy of environmental constraints on plant growth, this clustering in maximum stem height appears
to be stronger at ecoregions less limited by environmental conditions. As environments become harsher and less
optimal for plant growth, these clustering patterns disappear. This is particularly true for the severe climatic con-
ditions characteristic in the Mediterranean (with erratic rainfall, limited water availability and drought), as well as
of boreal zones (with low radiation incidence and cold temperatures). According to model predictions, the overall
clustering patterns found at middle-range latitudes are consistent with species competitive dominance controlling
species height differences.

**Two predictions from the explicit model tested against data**

The spatially-explicit model allows for either the dominance of tall, mid-sized or short plants, as a consequence
of the trade-off between investment in either potential growth or alternative mechanisms other than growth (see
Fig. 5 in Capitán *et al.* (2020)). We have tested whether taller or shorter plants are most commonly represented
in ecoregions via the correlation of cell-averaged heights and evapotranspiration (Fig. 6a), which shows a mixed
picture. With few exceptions, mid-latitude ecoregions exhibit positive correlation (taller plants are selected in
regions favoring plant growth), whereas negative dependencies are often observed in latitudinal extremes (Fig. 6b).
Correlations are significant but, in some cases, very weak. These results are consistent with our interpretation in
terms of a signal of competitive dominance in mid-latitude ecoregions.

Our spatially-explicit model predicts the persistence of trait clustering as species are aggregated at larger spatial
scales (much larger than the typical range of species interactions). This is important because real individual plants
interact at much lower spatial scales (1 to 1000ha) compared to the spatial resolution of our dataset (grid cell sizes
about 50 km). To assess the robustness of our results, we further investigated the effect of aggregation scales
on clustering patterns using plant data. In line with the ~~spatially-implicit~~ spatially-explicit model, the analysis
of herbaceous plant communities from mid-latitude ecoregions reveals that our results are robust to both up- and
down-scaling community sizes (see Fig 6c). Height clustering remains significant in a range of aggregated scales,
and extrapolates to smaller areas (under a random placement hypothesis, communities of smaller sizes were built
by randomly selecting a number of species as predicted by the empirical species-area relation, see Supporting
Information, section B). We conclude that clustering patterns at large scales is an emerging pattern that can be
interpreted as a signature of competitive dominance operating at much smaller spatial scales.

**Discussion**

In this work we have tested predictions from a model of species-rich interacting communities under dominant
competition (Capitán *et al.*, 2020). Our work is generally framed in modern coexistence theory (Chesson, 2000)~~and~~
, and inspired by the competition-similarity paradigm (Mayfield & Levine, 2010)~~using~~. We used macro-ecological
trait data at large spatial scales (Kunstler *et al.*, 2016) ~~-While to show that, while~~ potential evapotranspiration de-

[revised manuscript text omitted]

**References**

1.

Adler, P. B., Salguero-gómez, R., Compagnoni, A., Hsu, J. S., Ray-mukherjee, J., Adler, P. B., Salguero-gómez,
R., Compagnoni, A., Hsu, J. S. & Ray-mukherjee, J. (2014). Correction for Adler et al., Functional traits explain
variation in plant life history strategies. *Proceedings of the National Academy of Sciences*, 111, 10019–10019.

2.

Allesina, S. & Levine, J. M. (2011). A competitive network theory of species diversity. *Proc. Nat. Acad. Sci.*
*USA*, 108, 5638–5642.

3.

Alonso, D., Pinyol-Gallemí, A., Alcoverro, T. & Arthur, R. (2015). Fish community reassembly after coral mass
mortality: higher trophic groups are subject to increased rates of extinction. *Ecol. Lett.*, 18, 451–461.

4.

Buuren, S. & Groothuis-Oudshoorn, K. (2011). MICE: Multivariate imputation by chained equations in r. *J. Stat.*
*Softw.*, 45(3).

5.

Capitán, J. A., Cuenda, S. & Alonso, D. (2015). How similar can co-occurring species be in the presence of
competition and ecological drift? *J. R. Soc. Interface*, 12, 20150604.

6.

Capitán, J. A., Cuenda, S. & Alonso, D. (2017). Stochastic competitive exclusion leads to a cascade of species
extinctions. *J. Theor. Biol.*, 419, 137–151.

7.

Capitán, J. A., Cuenda, S. & Alonso, D. (2020). Competitive dominance in ecological communities: Modeling
approaches and theoretical predictions. *J. Theor. Biol.*, 502, 110349.

8.

Chase, J. M., Kraft, N. J. B., Smith, K. G., Vellend, M. & Inouye, B. D. (2011). Using null models to disentangle
variation in community dissimilarity from variation in α -diversity. *Ecosphere*, 2, 24.

9.

Chesson, P. L. (2000). Mechanisms of maintenance of species diversity. *Ann. Rev. Ecol. Syst.*, 31, 343–366.

10.

Craine, J. R. & Dybzinski, R. (2013). Mechanisms of plant competition for nutrients, water and light. *Funct.*
*Ecol.*, 27, 833–840.

- 11.
- Díaz, S., Kattge, J., Cornelissen, J. H. C., Wright, I. J., Lavorel, S., Dray, S., Reu, B., Kleyer, M., Wirth, C.,
Prentice, I. C., Garnier, E., Bönisch, G., Westoby, M., Poorter, H., Reich, P. B., Moles, A. T., Dickie, J., Gillison,
372 A. N., Zanne, A. E., Chave, J., Wright, S. J., Sheremet'ev, S. N., Jactel, H., Christopher, B., Cerabolini, B., Pierce,
S., Shipley, B., Kirkup, D., Casanoves, F., Joswig, J. S., Günther, A., Falczuk, V., Rüger, N., Mahecha, M. D. &
Gorné, L. D. (2015). The global spectrum of plant form and function. *Nature*, 529, 1–17.
- 12.
- Diniz-Filho, J. A. F., Rodríguez, M. Á., Bini, L. M., Olalla-Tarraga, M. Á., Cardillo, M., Nabout, J. C., Hortal,
377 J. & Hawkins, B. A. (2009). Climate history, human impacts and global body size of Carnivora (Mammalia:
Eutheria) at multiple evolutionary scales. *Journal of Biogeography*, 36, 2222–2236.
- 13.
- Falster, D. S. & Westoby, M. (2003). Plant height and evolutionary games. *Trends. Ecol. Evol.*, 18, 337–343.
- 14.
- Feng, G., Mi, X., Eiserhardt, W. L., Jin, G., Sang, W., Lu, Z., Wang, X., and B. Li, X. L., Sun, I., Ma, K. &
Svenning, J.-C. (2015). Assembly of forest communities across East-Asia. Insights from phylogenetic community
structure and species pool scaling. *Scientific Reports*, 5, 9337.
- 15.
- Garbulsky, M. F., Peñuelas, J., Papale, D., Ardö, J., Gouliden, M. L., Kiely, G., Richardson, A. D., Rotenberg,
E., Veenendaal, E. M. & Filella, I. (2010). Patterns and controls of the variability of radiation use efficiency and
primary productivity across terrestrial ecosystems. *Global Ecology and Biogeography*, 19, 253–267.
- 16.
- Gaudet, C. L. & Keddy, P. A. (1988). A comparative approach to predicting competitive ability from plants trait.
*Nature*, 334, 242–243.
- 17.
- Givnish, T. (1995). Plant stems: biomechanical adaptation for energy capture and influence on species distribu-
tions. In: *Plant stems: Physiology and Functional Morphology* (ed. Gartner, B.). Academic Press, Cambridge,
Massachusetts, pp. 3–49.
- 18.
- Givnish, T. J. (1982). Adaptive significance of leaf height in forest herbs. *Am. Nat.*, 112, 279–298.
- 19.
- Goldberg, D. E. & Barton, A. M. (1992). Patterns and Consequences of Interspecific Competition in Natural
Communities : A Review of Field Experiments with Plants. *The American naturalist*, 139, 771–801.

20.

Gotelli, N. J., Graves, G. R. & Rahbek, C. (2010). Macroecological signals of species interactions in the danish
avifauna. *Proc. Nat. Acad. Sci. USA*, 107, 5030–5035.

21.

Haegeman, B. & Loreau, M. (2011). A mathematical synthesis of niche and neutral theories in community
ecology. *J. Theor. Biol.*, 4, 263–271.

22.

Hart, S. P. & Marshall, D. J. (2013). Environmental stress, facilitation, competition, and coexistence. *Ecology*,
94, 2719–2731.

23.

HilleRisLambers, J., Adler, P., Harpole, W., Levine, J. & Mayfield, M. M. (2011). Rethinking Community As-
sembly Through the Lens of Coexistence Theory. *Annual Review of Ecology, Evolution, and Systematics*, 43,
120830113150004.

24.

Holmgren, M., Scheffer, M. & Huston, M. A. (1997). The interplay of facilitation and competition in plant
communities. *Ecology*, 78, 1966–1975.

25.

Jalas, J. & Suominen, J. (1964–1999). *Atlas Florae Europaeae. Distribution of vascular plants in Europe, Vol.*
*1–12*. Societas Biologica Fennica Vanamo, Helsinki.

26.

King, D. A. (1990). The adaptive significance of tree height. *Am. Nat.*, 135, 809–828.

27.

Kleyer, M., Bekker, R. M., Knevel, I. C., Bakker, J. P., Thompson, K., Sonnenschein, M., Poschod, P., Groe-
nendael, J. M. V., Klimes, L., Klimesová, J., Klotz, S., Rusch, G. M., Hermy, M., Adriaens, D., Boedeltje, G.,
Bossuyt, B., Dannemann, A., Endels, P., Götzenberger, L., Hodgson, J. G., Jackel, A.-K., Kühn, I., Kunzmann,
D., Ozinga, W. A., Römermann, C., Stadler, M., Schlegelmilch, J., Steendam, H. J., Tackenberg, O., Wilmann, B.,
Cornelissen, J. H. C., Eriksson, O., Garnier, E. & Peco, B. I. (2008). The leda traitbase: a database of life-history
traits of the northwest european flora. *Journal of Ecology*, 96, 1266–1274.

28.

Kraft, N. J. B., Godoy, O. & Levine, J. M. (2015). Plant functional traits and the multidimensional nature of
species coexistence. *Proceedings of the National Academy of Sciences*, 112, 797–802.

29.

Kunstler, G., Falster, D., Coomes, D. A., Hui, F., Kooyman, R. M., Laughlin, D. C., Poorter, L., Vanderwel,
434 M., Vieilledent, G., Wright, S. J., Aiba, M., Baraloto, C., Caspersen, J., Cornelissen, J. H. C., Gourlet-Fleury,
S., Hanewinkel, M., Herault, B., Kattge, J., Kurokawa, H., Onoda, Y., Peñuelas, J., Poorter, H., Uriarte, M.,
Richardson, S., Ruiz-Benito, P., Sun, I.-F., Stahl, G., Swenson, N. G., Thompson, J., Westerlund, B., and
437 M. A. Zavala, C. W., Zeng, H., Zimmerman, J. K., Zimmermann, N. E. & Westoby, M. (2016). Plant functional
traits have globally consistent effects on competition. *Nature*, 529, 204–207.

30.

Kunstler, G., Lavergne, S., Courbaud, B., and G. Vieilledent, W. T., Zimmermann, N. E., Kattge, J. & Coomes,
D. A. (2012). Competitive interactions between forest trees are driven by species' trait hierarchy, not phylogenetic
or functional similarity: implications for forest community assembly. *Ecol. Lett.*, 15, 831–840.

31.

Lambers, H. & Oliveira, R. S. (2020). *Plant Physiological Ecology*. Springer, Cham.

32.

Law, R., Marrow, P. & Dieckmann, U. (1997). On evolution under asymmetric competition. *Evol. Ecol.*, 11,
485–501.

33.

MacArthur, R. H. & Wilson, E. O. (1967). *The theory of island biogeography*. Princeton University Press,
Princeton.

34.

Maestre, F. T., Callaway, R. M., Valladares, F. & Lortie, C. J. (2009). Refining the stress-gradient hypothesis for
competition and facilitation in plant communities. *Journal of Ecology*, 97, 199–205.

35.

Mayfield, M. M. & Levine, J. M. (2010). Opposing effects of competitive exclusion on the phylogenetic structure
of communities. *Ecol. Lett.*, 13, 1085–1093.

36.

McKane, A. J., Alonso, D. & Solé, R. V. (2000). A mean field stochastic theory for species rich assembled
communities. *Phys. Rev. E*, 62, 8466–8484.

37.

Midgley, J. J. (2003). Is bigger better in plants? The hydraulic costs of increasing size in trees. *Trends. Ecol.*
*Evol.*, 18, 5–6.

38.

Mu, Q., Zhao, M. & Running, S. W. (2011). Improvements to a MODIS global terrestrial evapotranspiration
algorithm. *Remote Sens. Environ.*, 115, 1781–1800.

39.

Muller-Landau, H. (2010). The tolerance-fecundity trade-off and the maintenance of diversity in seed size. *Proc.*
*Nat. Acad. Sci. USA*, 107, 4242–4247.

40.

Olalla-Tárraga, M. Á. & Rodríguez, M. Á. (2007). Energy and interspecific body size patterns of amphibian
faunas in Europe and North America: Anurans follow Bergmann's rule, urodeles its converse. *Global Ecology*
*and Biogeography*, 16, 606–617.

41.

Olson, D. M., Dinerstein, E., Wikramanayake, E. D., Burgess, N. D., Powell, G. V. N., Underwood, E. C., D'amico,
475 J. A., Itoua, I., Strand, H. E., Morrison, J. C., Loucks, C. J., Allnutt, T. F., Ricketts, T. H., Kura, Y., Lamoreux,
476 J. F., Wettengel, W. W., Hedao, P. & Kassem, K. R. (2001). Terrestrial ecoregions of the world: A new map of life
on Earth. *BioScience*, 51, 993–938.

42.

Ordonez, A., Wright, I. J. & Olf, H. (2010). Functional differences between native and alien species: A global-
scale comparison. *Fun. Ecol.*, 24, 1353–1361.

43.

Pigolotti, S., López, C. & Hernández-García, E. (2007). Species clustering in competitive Lotka-Volterra models.
*Phys. Rev. Lett.*, 98, 1–4.

44.

Ricklefs, R. E. (2008). Disintegration of the ecological community. *Am. Nat.*, 172, 741–750.

45.

Ricklefs, R. E. (2011). Applying a regional community concept to forest birds of eastern North America. *Pro-*
*ceedings of the National Academy of Sciences of the United States of America*, 108, 2300–2305.

46.

Ricklefs, R. E. (2015). Intrinsic dynamics of the regional community. *Ecology Letters*, 18, 497–503.

47.

Ryan, M. G. & Yoder, B. J. (1997). Hydraulic limits to tree height and tree growth. *Bioscience*, 47, 235–242.

48.

Scheffer, M. & van Nes, E. H. (2006). Self-organized similarity, the evolutionary emergence of groups of similar
species. *Proceedings of the National Academy of Sciences of the United States of America*, 103, 6230–6235.

49.

Schoener, T. W. (1983). Field Experiments on Interspecific Competition. *The American naturalist*, 122, 240–285.

50.

Segura, A. M., Kruk, C., Calliari, D., García-Rodríguez, F., Conde, D., Widdicombe, C. E. & Fort, H. (2012).
Competition drives clumpy species coexistence in estuarine phytoplankton. *Scientific Reports*, 3, 1037.

51.

Serván, C. A., Capitán, J. A., Grilli, J., Morrison, K. E. & Allesina, S. (2018). Coexistence of many species in
random ecosystems. *Nature Ecology and Evolution*, 2, 1237.

52.

Soliveres, S., Maestre, F. T., Ulrich, W., Manning, P., Boch, S., Bowker, M. A., Prati, D., Delgado-Baquerizo,
506 M., Quero, J. L., Schöning, I., Gallardo, A., Weisser, W., Müller, J., Socher, S. A., García-Gómez, M., Ochoa, V.,
Schulze, E. D., Fischer, M. & Allan, E. (2015). Intransitive competition is widespread in plant communities and
maintains their species richness. *Ecology Letters*, 18, 790–798.

53.

Stanley Harpole, W. & Tilman, D. (2006). Non-neutral patterns of species abundance in grassland communities.
*Ecology Letters*, 9, 15–23.

54.

Stearns, S. C. (1989). Evolution in life-history Trade-offs. *Functional Ecology*, 3, 259–268.

55.

Stroud, J. T., Bush, M. R., Ladd, M. C., Nowicki, R. J., Shantz, A. A. & Sweatman, J. (2015). Is a community still
a community? Reviewing definitions of key terms in community ecology. *Ecology and Evolution*, 5, 4757–4765.

56.

Swenson, N. G., Enquist, B. J., Pither, J., Kerkhoff, A. J., Boyle, B., Weiser, M. D., Elser, J. J., Fagan, W. F.,
Forero-Montaña, J., Fyllas, N., Kraft, N. J. B., Lake, J. K., Moles, A. T., Patiño, S., Phillips, O. L., Price, C. A.,
Reich, P. B., Quesada, C. A., Stegen, J. C., Valencia, R., Wright, I. J., Wright, S. J., Andelman, S., Jorgensen,
P. M., Lacher, T. E., Monteagudo, A., Núñez-Vargas, M. P., Vasquez-Martínez, R. & Nolting, K. M. (2012). The
biogeography and filtering of woody plant functional diversity in North and South America. *Global Ecology and*
*Biogeography*, 21, 798–808.

57.

Tilman, D. (1982). *Resource Competition and Community Structure*. Princeton University Press, Princeton, NJ.

58.

Tilman, D. (1994). Competition and Biodiversity in Spatially Structured Habitats. *Ecology*, 75, 2–16.

59.

Tilman, D. (2004). Niche tradeoffs, neutrality, and community structure: A stochastic theory of resource compe-
tition, invasion, and community assembly. *Proc. Nat. Acad. Sci. USA*, 101, 10854–10861.

60.

Tilman, D. & Wedin, D. (1991). Plant Traits and Resource Reduction For Five Grasses Growing on a Nitrogen
Gradient. *Ecology*, 72, 685–700.

61.

Triadó-Margarit, X., Capitán, J. A., Menéndez-Serra, M., Ortiz-Álvarez, R., Ontiveros, V. J., Casamayor, E. O.
& Alonso, D. (2019). A Randomized Trait Community Clustering approach to unveil consistent environmental
thresholds in community assembly. *The ISME Journal*, 13.

62.

Valiente-Banuet, A. & Verdú, M. (2007). Assembly through facilitation can increase the phylogenetic diversity of
plant communities. *Ecol. Lett.*, 10, 1029–1036.

63.

Violle, C., Nemegut, D. R., Pu, Z. & Jiang, L. (2011). Phylogenetic limiting similarity and competitive exclusion.
*Ecol. Lett.*, 14, 782–787.

64.

Volkov, I., Banavar, J. R., Hubbell, S. P. & Maritan, A. (2009). Inferring species interactions in tropical forests.
*Proc. Nat. Acad. Sci. USA*, 106, 13854–13859.

65.

Wang, X. G., Wiegand, T., Kraft, N. J. B., Swenson, N. G., Davies, S. J., Hao, Z. Q., Howe, R., Lin, Y. C., Ma,

[revised manuscript text omitted]

mine effective competitive (biotic) interactions ~~among species in natural communities.~~ ~~The~~. According to this
framework, the balance between stabilizing trait differences and species dominance among competitors is crucial
to understand species coexistence ~~under this framework~~. In communities driven by fitness differences, species turn
out to be clustered around similar trait values selected through competitive dominance. However, trait clustering
may arise through two radically different mechanisms. Independent adaptation of non-interacting species to the
same environmental conditions can lead to trait clustering. The alternative explanation would say that competitive
interactions leading to fitness equalization end up producing more similar species, with, therefore, more similar
traits. Therefore, trait clustering may be interpreted as a fingerprint of competition even in the absence of envi-
ronmental filtering (Kraft *et al.*, 2015, Mayfield & Levine, 2010). ~~Community ecology, however, still needs~~ ~~These~~
ideas have been proved challenging to apply to large ecological communities. Rather than focusing on whether
(or not) and why ecological similarity among species should arise (or not) in natural communities, Hubbell and
colleagues assumed ecological equivalence as a first principle and studied the consequences of this assumption for
species coexistence and community-level patterns in species-rich systems (Alonso *et al.*, 2006, Hubbell, 2001, Rosindell *et al.*
. Other authors, building on the May's seminal work (1972), have used a random matrix approach to advance
understanding on species coexistence in large communities through mathematical analysis (Allesina & Grilli, 2020, Allesina
. Statistical physics has also helped to understand how pair-wise species interactions scale up to determine the type
of dynamic stability and potential species coexistence in species-rich large systems (Bunin, 2017).

Although the role of local interactions at determining large-scale diversity patterns is still controversial (Ricklefs, 2008)
, ~~community ecology lacks~~ a comprehensive theoretical framework able to ~~describe quantitatively~~ ~~explore quantitatively~~
~~to what extent~~ the role of biotic, species-to-species interactions ~~that are is~~ relevant to determine species composi-
tion and diversity across large spatial scales. Empirical studies, while they may be able to independently assess
environmental stress and species competitive abilities, are often limited to small community sizes (Violle *et al.*,
2011) or restricted to single habitats (Kunstler *et al.*, 2012). Very few studies have explored the idea of competi-
tion as a driver of community assembly across biogeographic regions (Kunstler *et al.*, 2016, Swenson *et al.*, 2012).
Here we attempted a continent-wide macro-ecological study of species assemblage patterns based on theoretic-
47 cal predictions from ~~modern coexistence theory~~ (Capitán *et al.*, 2020, Chesson, 2000, Mayfield & Levine, 2010)

a trait-driven theory of competitive dominance, based on extensions of a type of Lotka-Voleterra models. Our
theory applies to large ecological communities at large geographical scales –where species can be ranked in their
competitive ability according to certain species trait values (Capitán *et al.*, 2020).

Light and water availability (Fig. 1) impose significant limitations on gross primary productivity which is
reflected in actual evapotranspiration rates (Garbulsky *et al.*, 2010). These two resources vary at regional scales,
placing strong, sometimes opposing constraints on how tall a plant can grow ~~within the limits of structural stability~~.
Plant height is a fundamental trait that reflects the ability of the individual to optimize its own growth within its lo-
55 cal biotic environment and regional physical constraints (see Falster & Westoby (2003), Holmgren *et al.* (1997) and
56 references therein). How plant height adapts to these opposing constraints has been studied in trees (King, 1990,
Law *et al.*, 1997, Midgley, 2003) and herbaceous plants (Givnish, 1995, 1982). Here we analyzed presence-absence
matrices of floral herbaceous taxa across different European ecoregions to determine if competitive ability (re-
flected in maximum stem height) could help explain assemblage patterns at local scales across gradients of relevant
environmental factors such as evapotranspiration. We examined how well observed plant assemblages at macro-
ecological scales match theoretical predictions generated by a synthetic, stochastic framework of community as-
sembly (Capitán *et al.*, 2015, 2017, Haegeman & Loreau, 2011, McKane *et al.*, 2000)(Capitán *et al.*, 2015, 2017, Haegeman
, which we described in full detail in Capitán *et al.* (2020). By assuming that competition between hetero-specifics
is driven by signed height differences, we found a significant positive correlation between the degree of clustering
and actual evapotranspiration rates (or gross primary productivity, GPP). Across Europe, actual evapotranspiration
(and GPP) is lower at more southern latitudes (due to reduced precipitation levels) as well as at more northern
latitudes (due to colder temperatures and low levels of sunlight). Herbaceous plant height clustering is significant
only over a latitudinal band where environmental constraints to plant growth are weaker, which suggests that the
signature of competitive dominance can only be detected in the assemblage patterns of mid-latitude ecoregions.

**Theoretical predictions**

Recently, we presented a stochastic framework of community assembly (Capitán *et al.*, 2020). This framework
provides a stochastic extension of Lotka-Voleterra competition models. While other extensions consider only
symmetric competition on theoretical grounds (Haegeman & Loreau, 2011), our approach relates specifically measurable
species traits and competitive dominance. In order to make ~~to make~~ this contribution self-contained, we first pro-
vide a summary of the main predictions ~~derived by our suite of models –from our theory (Capitán *et al.*, 2020).~~ We
developed first a single-trait driven, spatially-implicit species-competition model. Then, we extended this model
into space and incorporated a second trait controlling species competition. Both models together provided us with
rich predictions that can be tested with appropriate species assembly data. Below we summarize these predictions.

Two predictions from the implicit model

Species coexistence decays with competition intensity

Recent theoretical approaches have focused on predicting analytically the expected fraction of species that survive
in competitive scenarios (Serván *et al.*, 2018). A spatially-implicit model of Lotka-Volterra type (Capitán *et al.*,
2020) allowed us to predict on average how many species are expected to survive as a function of mean competitive
strengths. We observed that the fraction of extant species p_c , which we called “coexistence probability”, decays
with the average competitive strength $\langle \rho \rangle$ as a power law above a certain threshold in competition, and curves for
different pool sizes S can be collapsed into the same curve following the mathematical dependence,

$$p_c \sim (\langle \rho \rangle S)^{-\gamma}, \quad (1)$$

which was observed numerically and justified analytically (see Capitán *et al.* (2020)). We showed that the exponent
γ is controlled by the immigration rate μ . This is the first prediction of the spatially implicit model.

Species clustering under competitive dominance

In order to explore the significance of competitive dominance in empirical communities, we applied first random-
ization tests to model communities. In this way, we established a second prediction for this model. Null models for
community assembly (Chase *et al.*, 2011, Gotelli *et al.*, 2010, Webb *et al.*, 2002) compare the properties of actual
communities against random samples of the same size extracted from a species pool (observed diversity at the
ecoregion level). This approach assumes that realized communities are built up through the independent arrival of
equivalent species from the pool (~~Alonso *et al.*, 2015, MacArthur & Wilson, 1967~~) (Alonso *et al.*, 2015, MacArthur & Wilson)
regardless of species preferences for particular environments or species interactions. Our randomization tests were
based on a single statistic, the competitive strength averaged over species present in realized model communities,
which were then compared to random samples of the same size drawn from the species pool. The null hypothesis
(i.e., empirical communities are built as random assemblages from the ecoregion) can be rejected in both sides of
the distribution, implying signals of ‘significant trait overdispersion’ (‘clustering’) if average trait differences are
larger (smaller) than expected at random. In the low immigration regime, the model predicts a significant signal
of clustering. This regime is characterized by a low non-dimensional immigration rate ($\lambda = \mu / (\alpha K)$ much lower
than 0) —here α stands for the average species growth rate in isolation, and K is the carrying capacity of the
environment.

Two predictions from the explicit model

The spatially-explicit model incorporates a trade-off between potential growth and ~~alternative mechanisms other~~
~~than growth that~~ the production of allelopathic compounds. This alternative mechanism would allow shorter

individuals to overcome being out-competed by taller plants (see Capitán *et al.* (2020)). ~~While the latter~~ Our
models explores how taller species, which are better competitors for light, ~~the former and shorter ones, which~~
allocate more energy in allelopathic compounds, coexist in a single interacting community on a given area (Fig. 1).

Competitive dominance may select for shorter plants

Height hierarchies alone, as assumed in our spatially-implicit model, lead to the selection of taller plants in species
assemblages. In the more realistic spatially-explicit model, species processes take place on a lattice where locally
taller plants grow faster than neighbors because they are less shaded, but in the presence of heterospecific neigh-
bors, they are also more prone to die. Computer simulations show that the balance of these two mechanisms can
end up selecting plant sizes characterized by an optimal potential height that can be either shifted toward lower
or higher values depending on the choice of model parameters. This is the first prediction of the spatially-explicit
model: species abundance distributions are not necessarily biased towards taller individuals, and they can peak at
species at intermediate or even shorter heights. In any case, and consistently, in this more complex scenario, a
balance between the gains of potential growth and the gains of energy allocation in allelopathy (as an example of a
non-size-related, alternative mechanism) may result in a selection for plants exhibiting significant height clustering
at stationarity.

Clustering patterns hold across aggregation scales

A second result that can be derived from the spatially-explicit model is related to the persistence of trait clustering
when species are aggregated over spatial scales larger than local interaction distances. Our spatially-explicit model
can help explain why clustering patterns persist over large scales. The distributions of species within a region may
reveal more information about the underlying assembly processes than the co-occurrence of species at any given
location (Ricklefs, 2008). As species are aggregated over lattice cells of increasing size, clustering patterns hold
even at scales much larger than local interaction distances. The model predicts consistent clustering patterns
regardless of the aggregation scale used to define species communities. This was the second prediction, derived
and carefully analyzed in Capitán *et al.* (2020), from our spatially-explicit model.

**Materials and methods**

Plant community data were drawn from Atlas Florae Europaeae (Jalas & Suominen, 1964–1999). The distribution
of flora is geographically described using equally-sized grid cells ($\sim 50 \times 50$ km) based on the Universal Transverse
Mercator projection and the Military Grid Reference System, see Fig. 2. Each cell was assigned to a dominant
habitat type based on the WWF Biomes of the World classification (Olson *et al.*, 2001), which defines different
ecoregions, i.e., geographically distinct assemblages of species subject to similar environmental conditions. We
consider each cell in an ecoregion to represent a species aggregation.

Each herbaceous species in an ecoregion was characterized by its maximum stem height H , an eco-morphological
 trait that relates to several critical functional strategies among plants (Díaz *et al.*, 2015). It represents an optimal
 trade-off between the gains of accessing light (King, 1990, Law *et al.*, 1997), water and nutrient transport from
 soil (Midgley, 2003, Ryan & Yoder, 1997), and additional constraints posed by the local biotic environment of
 each individual plant, such as competition, facilitation, or herbivory.

Mean height values were obtained from the LEDA database (Kleyer *et al.*, 2008) for as many species as there
 were available in the database. Missing values were taken from (Ordonez *et al.*, 2010) or inferred using a MICE
 (Multivariate Imputation by Chained Equations) approach (Buuren & Groothuis-Oudshoorn, 2011) together with
 a predictive mean matching algorithm based on other available traits (leaf and seed traits), genus, and growth
 forms as predictors. Based on plant growth forms, 2610 herbaceous species (aquatic, herbs, or graminoid) were
 considered in this work.

[revised manuscript text omitted]

Two predictions from the implicit model tested against data

Species coexistence decays with competition intensity

The collapse of curves predicted by Eq. (1) helps eliminate the variability in S , so that empirical coexistence
probabilities, which arise from different ecoregion sizes, can be fitted together (Fig. 3). Confirming the first
prediction of the spatially-implicit model, we found a significant correlation between the probability of coexistence
and the scaled competitive overlap based on empirical data (Fig. 3), indicating that a model driven solely by
dominant competitive interactions reliably predicts the average richness of plant communities across ecoregions.
In addition, this theoretical prediction allowed an indirect estimation of the relative importance $\hat{\rho}$ of average inter-
vs. intraspecific effects: the average ratio of inter- to intraspecific competition strength is about 5% (see Supporting
Information, section A for details on the estimation procedure).

Species clustering under competitive dominance

As a second prediction, the implicit model ~~implies high~~ predicts species clustering under competitive dominance
under certain parameter regime. High levels of trait clustering are only found for low immigration rates and
high carrying capacity values. Importantly, this ~~parameter regime is the parameter regime that seems to~~ precisely
emerges from the data. In Capitán *et al.* (2020) we derived a deterministic prediction for the exponent, $\gamma = 1$,
under no immigration, which does not match the one obtained from data ($\gamma = 0.61$). As ~~shown in that paper we~~
showed (Capitán *et al.*, 2020), it is a non-zero (but small) value of the immigration rate that determines the value
of the power-law exponent γ being lower than 1. that becomes lower than 1 in the case of non-zero immigration.
Indeed, for a realistic fit in Fig. 3, the exponent of the empirical power law is obtained for $\mu/\alpha \sim 0.1$ individuals
213 per generation. Since plant communities operate in a low-immigration regime, the non-dimensional immigration
rate $\lambda = \mu/(\alpha K)$ must satisfy $\lambda = 0.1/K \ll 1$, hence the carrying capacity must be large. ~~In a regime of~~ Indeed,
in the same parameter regime where empirical coexistence probabilities are best predicted, this is, low immigration
rate and high carrying capacity, ~~which best fits empirical coexistence probabilities,~~ the implicit model predicts a
significant degree of species clustering [see Fig. 3 in Capitán *et al.* (2020)].

~~Following Triadó-Margarit *et al.* (2019), our randomization tests applied to empirical communities were based~~
~~on the average competitive strength observed in a cell C formed by s species,~~

$$\langle \rho \rangle_C = \frac{2}{s(s-1)} \sum_{i=1}^s \sum_{j=i+1}^s |\rho_{ij}^C|,$$

~~where (ρ_{ij}^C) is the submatrix of the ecoregion competition matrix restricted to the species present in the cell.~~
~~Compared to ecoregion samples, the lower (higher) the empirical community average $\langle \rho \rangle_C$ is, the higher (lower) is~~
~~the degree of species clustering in the cell. For each cell we calculated the probability $p = \Pr(\langle \rho \rangle_Q \leq \langle \rho \rangle_C)$ that~~
~~the the competition average $\langle \rho \rangle_Q$ randomly sampled from the pool is smaller than the empirical average. At a 5%~~

significance level, if $p > 0.95$ the empirical competition average is significantly larger than the average measured
for random pool samples, which implies that average trait differences in realized communities are larger than would
be expected at random. On the other hand, if $p < 0.05$, observed trait differences are significantly smaller than
would be expected at random. Therefore, if $p > 0.95$, the community exhibits ‘significant trait overdispersion’;
whereas if $p < 0.05$, there is evidence for ‘significant trait clustering’ in the observed species assemblage.

Testing ~~the~~ this second prediction against empirical observations yields a mixed picture. We calculated p -
values for randomization tests applied to every cell in each ecoregion, which represent the empirical distribution
of p -values (Fig. 4). At the parameter values that make plant data consistent with the first prediction, the spatially-
implicit model predicts significant trait clustering. We observe that some ecoregions are consistent with this
theoretical expectation. However, other ecoregions clearly do not comply with this prediction. In addition, no
ecoregion is consistent with trait overdispersion (Fig. 4). Selecting species in randomization tests according to
species dispersal abilities portrays the same picture (results not shown).

Ecoregion clustering and actual evapotranspiration rates

~~In order to better quantify~~ We explored whether there is a geographic signal in the propensity of an ecoregion to
exhibit clustering in maximum stem height. For a better quantification, we defined a clustering index q for an
ecoregion as the fraction of its cells that lie within the 5% range of significant clustering (randomization tests yield
p -values smaller than 0.05 for those cells). An ecoregion for which significant clustering is found in most of its
cells will tend to score high in the q index. We examined how the clustering index varied across the continent
in terms of the geographical location of ecoregion centroids as well as with actual evapotranspiration (Fig. 5).

~~Evapotranspiration maps were obtained from data estimated through remote sensing (Mu *et al.*, 2011).~~

Water availability acts as a factor limiting plant growth at geographical scales (Fig. 1a); ~~and correlates with~~
~~gross primary productivity~~. However, water has to be channeled up through stems and leaves for effective
growth to take place. Therefore, at large geographic scales, growth primary productivity positively correlates
with evapotranspiration (Garbulsky *et al.*, 2010), see Fig. 5d. Therefore, for a given region, mean annual evapo-
transpiration is a reliable measure of environmental constraints on plant growth (Garbulsky *et al.*, 2010). Panels a
and b of Fig. 5 show a clear latitudinal trend: there is an intermediate range of ecoregion latitudes where both clus-
tering indices and evapotranspiration are large, indicating that evapotranspiration measures can robustly predict

[revised manuscript text omitted]

~~. In some of these studies, a trade-off between competitive and colonization abilities has particular trade-offs have~~
~~been shown to maintain plant diversity , although other hierarchies have been also suggested (Muller-Landau, 2010)~~
~~—and limiting similarity, which involves that competitive dominance may also lead to trait over-dispersion.~~
~~However, these theoretical results arise as a consequence of a particular tradeoff definition. We believe our~~
~~theoretical models are more general (Capitán *et al.*, 2020), and, in their diverse formulations, invariably lead to~~
~~the opposite pattern: trait clustering.~~ Interestingly, the relevant role of competitive dominance driven by species

trait hierarchies has been also reported at much smaller spatial scales for forest trees along an altitudinal gradient
in the French Alps (Kunstler *et al.*, 2012). Moreover, a ~~recent~~ study of the assembly of forest communities across
East Asia shows that a phylogenetic-based species similarity index tends to be smaller the higher the minimum
temperature of the coldest month is (Feng *et al.*, 2015). Although traits are not generally related to competitive
abilities, and they are diverse in their functionality and in their response to environmental stress, these studies,
together with our results, suggest that trait clustering is generally likely to occur where conditions for plant growth
are less restrictive. Our models indicate that the process underlying this pattern is competitive dominance rather
than Darwin's competition-similarity hypothesis, although it is likely that community assembly for other taxa may
be driven by other biotic or environmental filters. For instance, phytoplankton communities from estuarine ecosys-
tems (Segura *et al.*, 2012) are more consistent with Darwin's seminal hypothesis since they appear to be driven by
limiting similarity creating clumpy species coexistence (Pigolotti *et al.*, 2007, Scheffer & van Nes, 2006). Com-
petitive hierarchies are, of course, not hard-wired in nature. Intransitivities may still play a key role in maintaining
diversity in some systems (Allesina & Levine, 2011, Soliveres *et al.*, 2015, Zhang & Lamb, 2012).

In Capitán *et al.* (2020) we demonstrated how different coexistence *vs.* competition curves can be collapsed
into a single curve. Here we showed that model predictions were quantitatively consistent with the observed decay-
ing behavior of the probability of local coexistence as overall competition intensity increases. This general scaling
behavior is typical for stochastic community models in the presence of both symmetrical (Capitán *et al.*, 2015,
2017) and asymmetrical competition, as ~~presented here.~~ we showed in our previous publication (Capitán *et al.*, 2020)
. Here we tested this pattern at large geographical scales. The scaling allowed us to give a rough estimate of $\hat{\rho}$, an
average ratio of inter- *vs.* intraspecific competition (see Fig 3a). Our indirect method is only able to estimate an av-
erage $\hat{\rho}$ across ecoregions. This average estimate is a highly aggregated parameter calculated from the whole data
set, and therefore, characterizing European herbaceous plant communities. Although we expect high variability
in its value between ecoregions, in a given ecoregion, the ratio of inter- *vs.* intra-competition is expected to be, on
average, about 0.05. Whenever direct empirical estimates of the ratio of inter- *vs.* intra-competition are obtained, a
few similar species are typically studied using small-scale field experiments (Goldberg & Barton, 1992, Schoener,
1983). It is, therefore, unsurprising that empirical estimations of this parameter tend to be higher than ours (Kraft
*et al.*, 2015), but see also Volkov *et al.* (2009) and Wang *et al.* (2016). Being able to provide rough estimates of
this parameter at regional scales is also a novel result from our analysis. Our results are in agreement with a recent
study of trees across six forest biomes where the authors found that trait variation is mostly related to competitive
imbalances tending to drive inferior competitors to extinction (Kunstler *et al.*, 2016). Further work is required to
better relate the average ratio of inter- *vs.* intraspecific competition, which stabilizes species co-existence, to plant
traits, and analyze how this aggregated parameter changes at increasing spatial scales and across taxa.

In this paper we have explored several predictions from theoretical models aimed at describing plant dynam-
ics, which have been derived and carefully studied in Capitán *et al.* (2020). In total, we have contrasted four

model predictions against reported herbaceous plant diversity across Europe. ~~As usual, our~~ Our theoretical models
represent a strong over-simplification of real plant community dynamics. However, in spite of disregarding the
true complexity of these communities, our theory approach is useful, not only because it can reproduce macro-
ecological, observational patterns with a small number of meaningful aggregated variables, but also because it
provides new quantitative or qualitative predictions than may lead to new both empirical and observational studies.
We look forward to seeing our simple trait-driven theory of competitive dominance being falsified (or not) in other
ecological contexts. We humbly believe our message should be discussed within the context of the full scientific
community interested in biodiversity research. Finding a theoretically robust and ecologically meaningful rap-
prochement between theory and data at relevant scales remains a challenge for ecology, and we trust that our work
will inspire new contributions in this direction.

**Acknowledgments**

The authors thank Mercedes Pascual for her insightful comments, and are indebted to Rohan Arthur, Han Olf,
Joaquín Hortal, and Fernando Valladares for their constructive criticism on earlier versions of this manuscript. This
work was funded by the Spanish ‘Ministerio de Economía y Competitividad’ under the projects CGL2012-39964
and CGL2015-69043-P (DA, JAC), by the Spanish ‘Ministerio de Ciencia, Innovación y Universidades’ under the
project PGC2018-096577-B-I00 (DA, JAC), and the Ramón y Cajal Fellowship program (RYC-2010-06545, DA).
JAC acknowledges partial financial support from the Department of Applied Mathematics (Universidad Politécnica
de Madrid). SC acknowledges financial support from Banco Santander through grant PR87/19-22582.

**References**

1.

Adler, P. B., Salguero-gómez, R., Compagnoni, A., Hsu, J. S., Ray-mukherjee, J., Adler, P. B., Salguero-gómez,
R., Compagnoni, A., Hsu, J. S. & Ray-mukherjee, J. (2014). Correction for Adler et al., Functional traits explain
variation in plant life history strategies. *Proceedings of the National Academy of Sciences*, 111, 10019–10019.

2.

Allesina, S. & Grilli, J. (2020). Models for large ecological communities-a random matrix approach. In: *Theo-*
*retical Ecology: Concepts and Applications* (ed. McCann, KS and Gellner, G). Oxford University Press, USA.
ISBN 978-0-19-882428-2; 978-0-19-882429-9, pp. 74–92.

3.

Allesina, S. & Levine, J. M. (2011). A competitive network theory of species diversity. *Proc. Nat. Acad. Sci.*
*USA*, 108, 5638–5642.

- 4.
- Allesina, S. & Tang, S. (2012). Stability criteria for complex ecosystems. *Nature*, 483, 205–208.
- 5.
- Allesina, S. & Tang, S. (2015). The stability-complexity relationship at age 40: a random matrix perspective.
*Population Ecology*, 57, 63–75.
- 6.
- Alonso, D., Etienne, R. S. & McKane, A. J. (2006). The merits of neutral theory. *Trends Ecol. Evol.*, 21, 451–457.
- 7.
- Alonso, D., Pinyol-Gallemí, A., Alcoverro, T. & Arthur, R. (2015). Fish community reassembly after coral mass
mortality: higher trophic groups are subject to increased rates of extinction. *Ecol. Lett.*, 18, 451–461.
- 8.
- Bunin, G. (2017). Ecological communities with Lotka-Volterra dynamics. *PHYSICAL REVIEW E*, 95.
- 9.
- Buuren, S. & Groothuis-Oudshoorn, K. (2011). MICE: Multivariate imputation by chained equations in r. *J. Stat.*
*Softw.*, 45(3).
- 10.
- Capitán, J. A., Cuenda, S. & Alonso, D. (2015). How similar can co-occurring species be in the presence of
competition and ecological drift? *J. R. Soc. Interface*, 12, 20150604.
- 11.
- Capitán, J. A., Cuenda, S. & Alonso, D. (2017). Stochastic competitive exclusion leads to a cascade of species
extinctions. *J. Theor. Biol.*, 419, 137–151.
- 12.
- Capitán, J. A., Cuenda, S. & Alonso, D. (2020). Competitive dominance in ecological communities: Modeling
approaches and theoretical predictions. *J. Theor. Biol.*, 502, 110349.
- 13.
- Chase, J. M., Kraft, N. J. B., Smith, K. G., Vellend, M. & Inouye, B. D. (2011). Using null models to disentangle
variation in community dissimilarity from variation in α -diversity. *Ecosphere*, 2, 24.
- 14.
- Chesson, P. L. (2000). Mechanisms of maintenance of species diversity. *Ann. Rev. Ecol. Syst.*, 31, 343–366.

15.

Craine, J. R. & Dybzinski, R. (2013). Mechanisms of plant competition for nutrients, water and light. *Funct.*
*Ecol.*, 27, 833–840.

16.

Díaz, S., Kattge, J., Cornelissen, J. H. C., Wright, I. J., Lavorel, S., Dray, S., Reu, B., Kleyer, M., Wirth, C.,
Prentice, I. C., Garnier, E., Bönsch, G., Westoby, M., Poorter, H., Reich, P. B., Moles, A. T., Dickie, J., Gillison,
455 A. N., Zanne, A. E., Chave, J., Wright, S. J., Sheremet'ev, S. N., Jactel, H., Christopher, B., Cerabolini, B., Pierce,
S., Shipley, B., Kirkup, D., Casanoves, F., Joswig, J. S., Günther, A., Falczuk, V., Rüger, N., Mahecha, M. D. &
Gorné, L. D. (2015). The global spectrum of plant form and function. *Nature*, 529, 1–17.

17.

Diniz-Filho, J. A. F., Rodríguez, M. Á., Bini, L. M., Olalla-Tarraga, M. Á., Cardillo, M., Nabout, J. C., Hortal,
460 J. & Hawkins, B. A. (2009). Climate history, human impacts and global body size of Carnivora (Mammalia:
Eutheria) at multiple evolutionary scales. *Journal of Biogeography*, 36, 2222–2236.

18.

Ellner, S. P., Snyder, R. E., Adler, P. B. & Hooker, G. (2019). An expanded modern coexistence theory for
empirical applications. *Ecology Letters*, 22, 3–18.

19.

Falster, D. S. & Westoby, M. (2003). Plant height and evolutionary games. *Trends. Ecol. Evol.*, 18, 337–343.

20.

Feng, G., Mi, X., Eiserhardt, W. L., Jin, G., Sang, W., Lu, Z., Wang, X., and B. Li, X. L., Sun, I., Ma, K. &
Svenning, J.-C. (2015). Assembly of forest communities across East-Asia. Insights from phylogenetic community
structure and species pool scaling. *Scientific Reports*, 5, 9337.

21.

Garbulsky, M. F., Peñuelas, J., Papale, D., Ardö, J., Goulden, M. L., Kiely, G., Richardson, A. D., Rotenberg,
E., Veenendaal, E. M. & Filella, I. (2010). Patterns and controls of the variability of radiation use efficiency and
primary productivity across terrestrial ecosystems. *Global Ecology and Biogeography*, 19, 253–267.

22.

Gaudet, C. L. & Keddy, P. A. (1988). A comparative approach to predicting competitive ability from plants trait.
*Nature*, 334, 242–243.

23.

Givnish, T. (1995). Plant stems: biomechanical adaptation for energy capture and influence on species distribu-

- tions. In: *Plant stems: Physiology and Functional Morphology* (ed. Gartner, B.). Academic Press, Cambridge,
Massachusetts, pp. 3–49.
- 24.
- Givnish, T. J. (1982). Adaptive significance of leaf height in forest herbs. *Am. Nat.*, 112, 279–298.
- 25.
- Goldberg, D. E. & Barton, A. M. (1992). Patterns and Consequences of Interspecific Competition in Natural
Communities : A Review of Field Experiments with Plants. *The American naturalist*, 139, 771–801.
- 26.
- Gotelli, N. J., Graves, G. R. & Rahbek, C. (2010). Macroecological signals of species interactions in the danish
avifauna. *Proc. Nat. Acad. Sci. USA*, 107, 5030–5035.
- 27.
- Haegeman, B. & Loreau, M. (2011). A mathematical synthesis of niche and neutral theories in community
ecology. *J. Theor. Biol.*, 4, 263–271.
- 28.
- Hart, S. P. & Marshall, D. J. (2013). Environmental stress, facilitation, competition, and coexistence. *Ecology*,
94, 2719–2731.
- 29.
- HilleRisLambers, J., Adler, P., Harpole, W., Levine, J. & Mayfield, M. M. (2011). Rethinking Community As-
sembly Through the Lens of Coexistence Theory. *Annual Review of Ecology, Evolution, and Systematics*, 43,
120830113150004.
- 30.
- Holmgren, M., Scheffer, M. & Huston, M. A. (1997). The interplay of facilitation and competition in plant
communities. *Ecology*, 78, 1966–1975.
- 31.
- Hubbell, S. P. (2001). *The Unified Theory of Biodiversity and Biogeography*. Princeton University Press, Princeton.
- 32.
- Jalas, J. & Suominen, J. (1964–1999). *Atlas Florae Europaeae. Distribution of vascular plants in Europe, Vol.*
*1–12*. Societas Biologica Fennica Vanamo, Helsinki.
- 33.
- King, D. A. (1990). The adaptive significance of tree height. *Am. Nat.*, 135, 809–828.

34.

Kleyer, M., Bekker, R. M., Knevel, I. C., Bakker, J. P., Thompson, K., Sonnenschein, M., Poschlod, P., Groe-
nendael, J. M. V., Klimes, L., Klimesová, J., Klotz, S., Rusch, G. M., Hermy, M., Adriaens, D., Boedeltje, G.,
Bossuyt, B., Dannemann, A., Endels, P., Götzenberger, L., Hodgson, J. G., Jackel, A.-K., Kühn, I., Kunzmann,
D., Ozinga, W. A., Römermann, C., Stadler, M., Schlegelmilch, J., Steendam, H. J., Tackenberg, O., Wilmann, B.,
Cornelissen, J. H. C., Eriksson, O., Garnier, E. & Peco, B. I. (2008). The leda traitbase: a database of life-history
traits of the northwest european flora. *Journal of Ecology*, 96, 1266–1274.

35.

Kraft, N. J. B., Godoy, O. & Levine, J. M. (2015). Plant functional traits and the multidimensional nature of
species coexistence. *Proceedings of the National Academy of Sciences*, 112, 797–802.

36.

Kunstler, G., Falster, D., Coomes, D. A., Hui, F., Kooyman, R. M., Laughlin, D. C., Poorter, L., Vanderwel,
522 M., Vieilledent, G., Wright, S. J., Aiba, M., Baraloto, C., Caspersen, J., Cornelissen, J. H. C., Gourlet-Fleury,
S., Hanewinkel, M., Herault, B., Kattge, J., Kurokawa, H., Onoda, Y., Peñuelas, J., Poorter, H., Uriarte, M.,
Richardson, S., Ruiz-Benito, P., Sun, I.-F., Stahl, G., Swenson, N. G., Thompson, J., Westerlund, B., and
525 M. A. Zavala, C. W., Zeng, H., Zimmerman, J. K., Zimmermann, N. E. & Westoby, M. (2016). Plant functional
traits have globally consistent effects on competition. *Nature*, 529, 204–207.

37.

Kunstler, G., Lavergne, S., Courbaud, B., and G. Vieilledent, W. T., Zimmermann, N. E., Kattge, J. & Coomes,
D. A. (2012). Competitive interactions between forest trees are driven by species' trait hierarchy, not phylogenetic
or functional similarity: implications for forest community assembly. *Ecol. Lett.*, 15, 831–840.

38.

Lambers, H. & Oliveira, R. S. (2019). *Plant physiological ecology*. Springer.

39.

Law, R., Marrow, P. & Dieckmann, U. (1997). On evolution under asymmetric competition. *Evol. Ecol.*, 11,
485–501.

40.

Lotka, A. J. (1925). *Elements of Physical Biology*. Williams and Wilkins, Maryland, USA.

41.

MacArthur, R. H. & Wilson, E. O. (1967). *The theory of island biogeography*. Princeton University Press,
Princeton.

42.

Maestre, F. T., Callaway, R. M., Valladares, F. & Lortie, C. J. (2009). Refining the stress-gradient hypothesis for
competition and facilitation in plant communities. *Journal of Ecology*, 97, 199–205.

43.

May, R. M. (1972). Will a large complex system be stable? *Nature*, 238, 413–414.

44.

Mayfield, M. M. & Levine, J. M. (2010). Opposing effects of competitive exclusion on the phylogenetic structure
of communities. *Ecol. Lett.*, 13, 1085–1093.

45.

McKane, A. J., Alonso, D. & Solé, R. V. (2000). A mean field stochastic theory for species rich assembled
communities. *Phys. Rev. E*, 62, 8466–8484.

46.

Midgley, J. J. (2003). Is bigger better in plants? The hydraulic costs of increasing size in trees. *Trends. Ecol.*
*Evol.*, 18, 5–6.

47.

Mu, Q., Zhao, M. & Running, S. W. (2011). Improvements to a MODIS global terrestrial evapotranspiration
algorithm. *Remote Sens. Environ.*, 115, 1781–1800.

48.

Muller-Landau, H. (2010). The tolerance-fecundity trade-off and the maintenance of diversity in seed size. *Proc.*
*Nat. Acad. Sci. USA*, 107, 4242–4247.

49.

Olalla-Tárraga, M. Á. & Rodríguez, M. Á. (2007). Energy and interspecific body size patterns of amphibian
faunas in Europe and North America: Anurans follow Bergmann's rule, urodeles its converse. *Global Ecology*
*and Biogeography*, 16, 606–617.

50.

Olson, D. M., Dinerstein, E., Wikramanayake, E. D., Burgess, N. D., Powell, G. V. N., Underwood, E. C., D'amico,
567 J. A., Itoua, I., Strand, H. E., Morrison, J. C., Loucks, C. J., Allnutt, T. F., Ricketts, T. H., Kura, Y., Lamoreux,
568 J. F., Wettengel, W. W., Hedao, P. & Kassem, K. R. (2001). Terrestrial ecoregions of the world: A new map of life
on Earth. *BioScience*, 51, 993–938.

51.

Ontiveros, V. J., Capitán, J. A., Arthur, R., Casamayor, E. O. & Alonso, D. (2019). Colonization and extinction

rates estimated from temporal dynamics of ecological communities: The island r package. *Methods in Ecology*
*and Evolution*, 10, 1108–1117.

52.

Ordonez, A., Wright, I. J. & Olf, H. (2010). Functional differences between native and alien species: A global-
scale comparison. *Fun. Ecol.*, 24, 1353–1361.

53.

Pigolotti, S., López, C. & Hernández-García, E. (2007). Species clustering in competitive Lotka-Volterra models.
*Phys. Rev. Lett.*, 98, 1–4.

54.

Ricklefs, R. E. (2008). Disintegration of the ecological community. *Am. Nat.*, 172, 741–750.

55.

Ricklefs, R. E. (2011). Applying a regional community concept to forest birds of eastern North America. *Pro-*
*ceedings of the National Academy of Sciences of the United States of America*, 108, 2300–2305.

56.

Ricklefs, R. E. (2015). Intrinsic dynamics of the regional community. *Ecology Letters*, 18, 497–503.

57.

Rosindell, J., Hubbell, S. P. & Etienne, R. S. (2011). The unified neutral theory of biodiversity and biogeography
at age ten. *Trends Ecol. Evol.*, 26, 451–457.

58.

Ryan, M. G. & Yoder, B. J. (1997). Hydraulic limits to tree height and tree growth. *Bioscience*, 47, 235–242.

59.

Scheffer, M. & van Nes, E. H. (2006). Self-organized similarity, the evolutionary emergence of groups of similar
species. *Proceedings of the National Academy of Sciences of the United States of America*, 103, 6230–6235.

60.

Schoener, T. W. (1983). Field Experiments on Interspecific Competition. *The American naturalist*, 122, 240–285.

61.

Segura, A. M., Kruk, C., Calliari, D., García-Rodríguez, F., Conde, D., Widdicombe, C. E. & Fort, H. (2012).
Competition drives clumpy species coexistence in estuarine phytoplankton. *Scientific Reports*, 3, 1037.

62.

Serván, C. A., Capitán, J. A., Grilli, J., Morrison, K. E. & Allesina, S. (2018). Coexistence of many species in
random ecosystems. *Nature Ecology and Evolution*, 2, 1237.

63.

Solé, R. V., Alonso, D. & McKane, A. J. (2000). Scaling in a network model of multispecies communities.
*Physica A*, 286, 337–344.

64.

Soliveres, S., Maestre, F. T., Ulrich, W., Manning, P., Boch, S., Bowker, M. A., Prati, D., Delgado-Baquerizo,
608 M., Quero, J. L., Schöning, I., Gallardo, A., Weisser, W., Müller, J., Socher, S. A., García-Gómez, M., Ochoa, V.,
Schulze, E. D., Fischer, M. & Allan, E. (2015). Intransitive competition is widespread in plant communities and
maintains their species richness. *Ecology Letters*, 18, 790–798.

65.

Stanley Harpole, W. & Tilman, D. (2006). Non-neutral patterns of species abundance in grassland communities.
*Ecology Letters*, 9, 15–23.

66.

Stearns, S. C. (1989). Evolution in life-history Trade-offs. *Functional Ecology*, 3, 259–268.

67.

Stroud, J. T., Bush, M. R., Ladd, M. C., Nowicki, R. J., Shantz, A. A. & Sweatman, J. (2015). Is a community still
a community? Reviewing definitions of key terms in community ecology. *Ecology and Evolution*, 5, 4757–4765.

68.

Swenson, N. G., Enquist, B. J., Pither, J., Kerkhoff, A. J., Boyle, B., Weiser, M. D., Elser, J. J., Fagan, W. F.,
Forero-Montaña, J., Fyllas, N., Kraft, N. J. B., Lake, J. K., Moles, A. T., Patiño, S., Phillips, O. L., Price, C. A.,
Reich, P. B., Quesada, C. A., Stegen, J. C., Valencia, R., Wright, I. J., Wright, S. J., Andelman, S., Jorgensen,
P. M., Lacher, T. E., Monteagudo, A., Núñez-Vargas, M. P., Vasquez-Martínez, R. & Nolting, K. M. (2012). The
biogeography and filtering of woody plant functional diversity in North and South America. *Global Ecology and*
*Biogeography*, 21, 798–808.

69.

Tilman, D. (1982). *Resource Competition and Community Structure*. Princeton University Press, Princeton, NJ.

70.

Tilman, D. (1994). Competition and Biodiversity in Spatially Structured Habitats. *Ecology*, 75, 2–16.

71.

Tilman, D. (2004). Niche tradeoffs, neutrality, and community structure: A stochastic theory of resource compe-
tition, invasion, and community assembly. *Proc. Nat. Acad. Sci. USA*, 101, 10854–10861.

72.

Tilman, D. & Wedin, D. (1991). Plant Traits and Resource Reduction For Five Grasses Growing on a Nitrogen
Gradient. *Ecology*, 72, 685–700.

73.

Triadó-Margarit, X., Capitán, J. A., Menéndez-Serra, M., Ortiz-Álvarez, R., Ontiveros, V. J., Casamayor, E. O.
& Alonso, D. (2019). A Randomized Trait Community Clustering approach to unveil consistent environmental
thresholds in community assembly. *The ISME Journal*, 13.

74.

Valiente-Banuet, A. & Verdú, M. (2007). Assembly through facilitation can increase the phylogenetic diversity of
plant communities. *Ecol. Lett.*, 10, 1029–1036.

75.

Violle, C., Nemegut, D. R., Pu, Z. & Jiang, L. (2011). Phylogenetic limiting similarity and competitive exclusion.
*Ecol. Lett.*, 14, 782–787.

76.

Volkov, I., Banavar, J. R., Hubbell, S. P. & Maritan, A. (2009). Inferring species interactions in tropical forests.
*Proc. Nat. Acad. Sci. USA*, 106, 13854–13859.

77.

Volterra, V. (1926). Variazioni e fluttuazioni del numero d'individui in specie animali conviventi. *Mem. Acad.*
*Naz. Lincei*, 2, 31–113.

78.

Wang, X. G., Wiegand, T., Kraft, N. J. B., Swenson, N. G., Davies, S. J., Hao, Z. Q., Howe, R., Lin, Y. C., Ma,
654 K. P., Mi, X. C., Su, S. H., Sun, I. F. & Wolf, A. (2016). Stochastic dilution effects weaken deterministic effects
of niche-based processes in species rich forests. *Ecology*, 97, 347–360.

79.

Webb, C. O., Ackerly, D. D., McPeck, M. A. & Donoghue, M. J. (2002). Phylogenies and community ecology.
*Ann. Rev. Ecol. Syst.*, 33, 475–505.

80.

Weiner, J. (1993). Competition among plants. *Treballs de la Societat Catalana de Biologia*, 44, 99–109.

81.

Zhang, S. & Lamb, E. G. (2012). Plant competitive ability and the transitivity of competitive hierarchies change
with plant age. *Plant Ecology*, 213, 15–23.

[revised manuscript text omitted]

Figure 2

Figure 3

Figure 4

Figure 5

Figure 6